Osteology of the Late Triassic aetosaur Scutarx deltatylus (Archosauria: Pseudosuchia)

Parker William G. 1 2 William_Parker@nps.gov
1 Division of Resource Management, Petrified Forest National Park , Petrified Forest, Arizona , United States
2 Jackson School Museum of Earth History, University of Texas at Austin , Austin, Texas , United States
Young Mark
Electronic publication date: 2016 Aug 30
Publication date: 2016
Volume: 4
Electronic Location ID: e2411
Received 2016 Apr 3; Accepted 2016 Aug 4
Copyright year: 2016
License: This is an open access article, free of all copyright, made available under the Creative Commons Public Domain Dedication. This work may be freely reproduced, distributed, transmitted, modified, built upon, or otherwise used by anyone for any lawful purpose.
License URL: https://creativecommons.org/publicdomain/zero/1.0/

Keywords: Late Triassic, Chinle Formation, Aetosauria, Petrified Forest, Archosauria, Biostratigraphy

Funding: Financial assistance for this project was provided by the Jackson School of Geosciences, the National Park Service and the Petrified Forest Museum Association. The funders had no role in study design, data collection and analysis, decision to publish, or preparation of the manuscript.

==============================
Aetosaurians are some of the most common fossils collected from the Upper Triassic Chinle Formation of Arizona, especially at the Petrified Forest National Park (PEFO). Aetosaurians collected from lower levels of the park include Desmatosuchus spurensis, Paratypothorax, Adamanasuchus eisenhardtae, Calyptosuchus wellesi, and Scutarx deltatylus. Four partial skeletons collected from the park between 2002 and 2009 represent the holotype and referred specimens of Scutarx deltatylus. These specimens include much of the carapace, as well as the vertebral column, and shoulder and pelvic girdles, and a new naming convention proposed for osteoderms descriptions better differentiates portions of the carapace and ventral armor. A partial skull from the holotype specimen represents the first aetosaur skull recovered and described from Arizona since the 1930s. The key morphological feature distinguishing Scutarx deltatylus is the presence of a prominent, triangular boss located in the posteromedial corner of the dorsal surface of the dorsal paramedian osteoderms. Scutarx deltatylus can be distinguished from closely related forms Calyptosuchus wellesi and Adamanasuchus eisenhardtae not only morphologically, but also stratigraphically. Thus, Scutarx deltatylus is potentially an index taxon for the upper part of the Adamanian biozone.

Introduction

The Triassic Period is a key transitional point in Earth history when remnants of Paleozoic terrestrial biotas were replaced by a Mesozoic biota including components of recent ecosystems (e.g., Fraser, 2006). Prominent in this new radiation were the archosaurs, which include the common ancestor of birds and crocodylians and all of their descendants (Gauthier, 1986). The early appearance and diversification of this important clade is of interest because, beginning in the Triassic, the archosaurs almost completely dominated all continental ecosystems throughout the entire Mesozoic (e.g., Nesbitt, 2011). Because the Triassic globe had a coalesced supercontinent, Pangaea, the Laurasian and Gondwanan continental faunas are often considered to be cosmopolitan in their distribution, presumably because of a lack of major oceanic barriers (Colbert, 1971). Thus, many Triassic taxa have been considered widespread and widely applicable for global biostratigraphy (e.g., Lucas, 1998).

More recent work suggests that this is a gross oversimplification of the taxonomic diversity present at the time (e.g., Irmis et al., 2007a; Nesbitt, Irmis & Parker, 2007; Nesbitt et al., 2009a; Nesbitt et al., 2009b) and new research on many Triassic groups is showing evidence for endemism of species-level taxa (e.g., Martz & Small, 2006; Parker, 2008a; Parker, 2008b; Stocker, 2010), with distinct patterns of radiation of more inclusive clades into new areas (e.g., Nesbitt et al., 2010). Key to this change in thinking are the utilization of testable techniques such as apomorphy-based identification of fossils (Irmis et al., 2007b; Nesbitt & Stocker, 2008) and improved phylogenetic approaches to archosaur relationships and paleobiogeography (e.g., Irmis, 2008; Nesbitt, 2011; Nesbitt et al., 2010). The apomorphy-based approach reveals hidden diversity in faunal assemblages resulting in the recognition of distinct taxa (Nesbitt & Stocker, 2008).

Aetosaurians are quadrupedal, heavily armored, suchian archosaurs with a global distribution, restricted to non-marine strata of the Late Triassic (Desojo et al., 2013). Aetosaurians are characterized by their specialized skull with partially edentulous mandibles, an upturned premaxillary tip, and laterally facing supratemporal fenestrae. Another key feature of aetosaurians is a heavy carapace consisting of four columns of rectangular dermal armor, two paramedian columns that straddle the midline, and two lateral columns (Walker, 1961). Ventral and appendicular osteoderms are also present in most taxa. Aetosaurian osteoderms possess detailed ornamentation on the dorsal surface, the patterning of which can be diagnostic for taxa (Long & Ballew, 1985). Thus, the type specimens of several aetosaurian taxa consist solely of osteoderms (e.g., Typothorax coccinarum Cope, 1875; Paratypothorax andressorum Long & Ballew, 1985; Lucasuchus hunti Long & Murry, 1995; Rioarribasuchus chamaensis Zeigler, Heckert & Lucas, 2002; Apachesuchus heckerti Spielmann & Lucas, 2012) or consist chiefly of osteoderms (e.g., Calyptosuchus wellesi Long & Ballew, 1985; Typothorax antiquus Lucas, Heckert & Hunt, 2003; Tecovasuchus chatterjeei Martz & Small, 2006; Adamanasuchus eisenhardtae Lucas, Hunt & Spielmann, 2007; Sierritasuchus macalpini Parker, Stocker & Irmis, 2008). Aetosaurian osteoderms and osteoderm fragments are among the most commonly recovered fossils from Upper Triassic strata (Heckert & Lucas, 2000). Because of this abundance, in concert with the apparent ease of taxonomic identification, global distribution in non-marine strata, and limited stratigraphic range (e.g., Upper Triassic), aetosaurians have been proposed as key index fossils for use in regional and global non-marine biostratigraphy (Long & Ballew, 1985; Lucas & Hunt, 1993; Lucas & Heckert, 1996; Lucas et al., 1997; Lucas, 1998; Heckert et al., 2007a; Heckert et al., 2007b; Lucas et al., 2007; Parker & Martz, 2011). Four Land Vertebrate Faunachrons (LVF) were erected that use aetosaurians to divide the Late Triassic Epoch (Lucas & Hunt, 1993); from oldest to youngest, these are the Otischalkian (middle Carnian); Adamanian (late Carnian); Revueltian (Norian), and the Apachean (Rhaetian). These were redefined as biozones by Parker & Martz (2011).

Aetosaurians are one of the most commonly recovered vertebrate fossils in the Upper Triassic Chinle Formation at Petrified Forest National Park (PEFO), Arizona. Paleontological investigations in the park between 2001 and 2009 resulted in the discovery of four partial skeletons that are considered a new taxon (Parker, 2016). The first specimen (PEFO 31217), discovered in 2001 and collected in 2002 from Petrified Forest Vertebrate Locality (PFV) 169 (Battleship NW Quarry; Fig. 1), was initially assigned to Calyptosuchus (= Stagonolepis) wellesi based on characters of the armor and vertebrae (Parker & Irmis, 2005). The second partial skeleton was collected in 2004 from PFV 304 (Milkshake Quarry), at the south end of the park (Fig. 1). That specimen (PEFO 34045) was also mentioned by Parker & Irmis (2005), who noted differences in the armor from Calyptosuchus wellesi and suggested that might represent a distinct species. The other two specimens were collected in 2007 and 2009. The first (PEFO 34616), from the Billings Gap area (PFV 355; Fig. 1) is notable because it included the first aetosaurian skull to be recovered in the park. The second specimen (PEFO 34919) was recovered from the Saurian Valley area of the Devils Playground (PFV 224; Fig. 1). All four of these specimens were originally assigned to Calyptosuchus wellesi by Parker & Martz (2011) and used to construct the stratigraphic range for that taxon. Calyptosuchus is considered to be an index taxon of the Adamanian biozone (Lucas & Hunt, 1993; Parker & Martz, 2011).

Figure 1 Map of Petrified Forest National Park showing relevant vertebrate fossil localities.

Modified from Parker & Irmis (2005).

Subsequent preparation and more detailed examination of these four specimens led to the discovery that they all shared a key autapomorphy, the presence of a prominent, raised triangular protuberance in the posteromedial corner of the paramedian osteoderms. The protuberance is not present on any of the osteoderms of the holotype of Calyptosuchus wellesi (UMMP 13950). It is also absent on the numerous paramedian osteoderms of Calyptosuchus wellesi recovered from the Placerias Quarry of Arizona in collections at the UCMP and the MNA. That autapomorphy and several features of the cranium and pelvis differentiate these specimens (PEFO 31217; PEFO 34045; PEFO 34616; PEFO 34919) from all other known aetosaurians and form the basis for assigning these materials to a new taxon, Scutarx deltatylus (Parker, 2016). The goal of this contribution is to provide a detailed osteological description of the holotype and paratype material of Scutarx deltatylus, and to discuss the potential biostratigraphic utility of the taxon locally and regionally.

Geological setting

The four localities from which the material of Scutarx deltatylus was collected all occur in the lower part of the Sonsela Member of the Chinle Formation (Martz & Parker, 2010) (Fig. 2). In the PEFO region the Sonsela Member can be divided into five distinct units, the Camp Butte, Lot’s Wife, Jasper Forest, Jim Camp Wash, and Martha’s Butte beds (Martz & Parker, 2010). The Lot’s Wife, Jasper Forest, and Martha’s Butte beds are sandstone dominated, cliff forming units with source areas to the south and west (Howell & Blakey, 2013), whereas the Lot’s Wife and Martha’s Butte beds are slope forming units with a higher proportion of mudrocks than sandstones (Martz & Parker, 2010). All of these localities represent proximal floodplain facies associated with a braided river system (Howell & Blakey, 2013; Martz & Parker, 2010; Woody, 2006).

Figure 2 Regional stratigraphy of the Petrified Forest area showing the stratigraphic position of the localities discussed in the text.

All occurrences are in the lower part of the Sonsela Member of the Chinle Formation and are within the Adamanian biozone. Stratigraphy from Martz & Parker (2010). Biozones from Parker & Martz (2011) and Reichgelt et al. (2013). Ages from Ramezani et al. (2011) and Atchley et al. (2013).

PFV 169 and PFV 224 occur in the upper part of the Lot’s Wife beds, PFV 355 is situated in the base of the Jasper Forest bed, and PFV 304 marks the highest stratigraphic occurrence, located in the lower part of the Jim Camp Wash beds (Fig. 2). All of these sites are below the ‘persistent red silcrete,’ a thick, chert, marker bed that approximates the stratigraphic boundary between the Adamanian and Revueltian biozones (Martz & Parker, 2010; Parker & Martz, 2011). Exact locality information is available at PEFO to qualified researchers. Non-disclosure of locality information is protected by the Paleontological Resources Preservation Act of 2009.

A high concentration of volcanic material in mudrocks of the Chinle Formation includes detrital zircons and allows for determination of high precision radioisotopic dates for studied beds (Fig. 2; Ramezani et al., 2011). Zircons from the top of the Lot’s Wife beds provided an age of 219.317 ± 0.080 Ma (sample SBJ; Ramezani et al., 2011). The base of the unit is constrained by a maximum depositional age of 223.036 ± 0.059 Ma for the top of the underlying Blue Mesa Member (sample TPs; Ramezani et al., 2011). Maximum depositional ages of 218.017 ± 0.088 Ma (sample GPL) and 213.870 ± 0.078 (sample KWI) are known from the Jasper Forest bed and the overlying Jim Camp Wash beds further constraining the upper age for the fossil specimens (Ramezani et al., 2011).

Materials and Methods

All specimens were excavated utilizing small hand tools, although a backhoe tractor was used initially to remove overburden at PFV 304. B-15 Polyvinyl Acetate “Vinac” (Air Products & Chemicals, Inc.) and B-76 Butvar (Eastman Chemical Company) dissolved in acetone were used as a consolidant in the field. PEFO 31217 was discovered partly in unconsolidated, heavily weathered sediment with numerous plant roots growing over and through the bones. Small hand tools, including brushes, caused damage to the bone surface so plastic drinking straws were used to blow away sediment from the bone surface, which was then quickly hardened with a consolidant. In the lab the same specimen (PEFO 31217) quickly deteriorated upon exposure and applications of Polyvinyl Acetate (Vinac™; Air Products and Chemicals, Inc.) proved to result in a flexible specimen, therefore liberal amounts of extremely thin Paleobond™ Penetrant Stabilizer PB002 (Uncommon Conglomerates) were applied to stop disintegration and provide rigidity of the bone. Because of the delicate nature of this specimen and the application of the cyanoacrylate, many of the bones cannot be prepared further or removed from the original field jackets. Furthermore, during collection the condition of the bones and surrounding matrix proved to be so poor that a portion of the jacket with the scapulocoracoid in it was lost during turning. This lost material consisted mostly of trunk vertebrae, ribs, and osteoderms. Unfortunately, this block of material is too large to CT scan to obtain more information for these elements.

The other three skeletons (PEFO 34045; PEFO 34616; PEFO 34919) were consolidated in the lab using B-72 Butvar™ (Eastman Chemical Company), with Paleobond™ PB40 and PB100 (Uncommon Conglomerates) cyanoacrylate used in many cases for permanent bonds. Paleobond™ PB304 (Uncommon Conglomerates) aerosol activator was originally used on some of the bones in PEFO 34045, but was halted because it was causing discoloration of the bone surface during the curing process. PEFO 34919 is coated with thin layers of hematite as is common for fossil specimens recovered from sandy facies in the Devils’ Playground region of PEFO. Mechanical preparation with pneumatic tools damaged the bone surface upon removing the coating and revealed that the hematite had permeated numerous microfractures in the bones, expanding them slightly, or in some bones significantly. As a result, the non-osteoderm bones from PFV 224 are highly deformed and often ‘mashed’ into the associated osteoderms. Further preparation to remove the hematite coating was not attempted.

Naming conventions for aetosaurian osteoderms

Traditionally, identification and naming of aetosaurian osteoderms, which cover the dorsal, ventral, and appendicular areas, utilizes terms first originated by Long & Ballew (1985). In this convention the dorsal armor (carapace) consists of two midline ‘paramedian’ columns flanked laterally by two ‘lateral’ columns (Long & Ballew, 1985; Heckert & Lucas, 1999; Desojo et al., 2013). By convention, osteoderms of the dorsal region are named from the type of vertebrae they cover (e.g., cervical, dorsal, and caudal; Long & Ballew, 1985). However, the anteriormost paramedian osteoderms lack equivalent lateral osteoderms causing a potential numbering offset between the presacral paramedian and lateral rows (Heckert et al., 2010). Aetosaurians also possess ventral armor at the throat, as well as ventral armor that underlies the ‘dorsal’ (= trunk) and caudal vertebrae. The presence of ventral armor of the ‘dorsal’ series creates the awkward combination of ‘ventral-dorsal’ osteoderms. Therefore, there is a need to standardize the positional nomenclature for aetosaurian osteoderms.

The term carapace properly refers only to the dorsally situated network of osteoderms, thus the term ‘dorsal carapace’ is incorrect and redundant. In this study, the term carapace refers only to the dorsally situated osteoderms and the term ventral osteoderms is used for all ventrally situated osteoderms (Heckert & Lucas, 1999).

The carapace can be divided into four anteroposteriorly trending columns of osteoderms (Heckert & Lucas, 1999; Heckert et al., 2010). Those that straddle the mid-line are referred to as the paramedians and the flanking osteoderms are called the lateral armor (Long & Ballew, 1985). Each column is divided into rows (Fig. 3) and as noted above these have traditionally been given names based on the vertebral series they cover (in most taxa there is a 1:1 ratio between osteoderms and vertebrae, except in the cervical series of desmatosuchines where six osteoderms cover the nine cervical vertebrae).

Figure 3 Differention and terminology for aetosaurian osteoderms, based on Stagonolepis robertsoni.

Reconstruction courtesy of Jeffrey Martz.

The two anteriormost paramedian osteoderms fit into the back of the skull and are generally mediolaterally oval and lack corresponding lateral osteoderms. These osteoderms are termed the nuchal series (Fig. 3; Sawin, 1947; Desojo et al., 2013; Schoch & Desojo, 2016). Posterior to these are roughly five, six, or nine rows of paramedian and lateral osteoderms that cover the entire cervical vertebral series, termed cervical osteoderms (Fig. 3; Long & Ballew, 1985). The patch of osteoderms beneath the cervical vertebrae in the throat area would be called the gular osteoderms, based on the name given to these osteoderms in phytosaurians (Long & Murry, 1995).

The next vertebral series initiates with the 10th presacral vertebra. On this vertebra the parapophysis has moved up to the top of the centrum, just below the level of the neurocentral suture. In the previous nine vertebrae (the cervical series), the parapophysis is situated at the base of the centrum, and in the eleventh vertebra the parapophysis is situated on the transverse process. Thus, the 10th presacral is transitional in form and has been considered to be the first of the ‘dorsal’ series (Case, 1922; Walker, 1961; Parker, 2008a), and that convention is followed here.

Historically in aetosaurians these vertebrae have been referred to as the dorsal series and osteoderms covering these vertebrae are the ‘dorsal osteoderms’ (e.g., Long & Ballew, 1985; Long & Murry, 1995; Heckert & Lucas, 2000; Desojo et al., 2013); however, this term has become problematic because whereas all of the osteoderms below the vertebral column are termed the ventral osteoderms, only those osteoderms above the vertebral column in the trunk region are called the dorsals. Thus, technically the osteoderms beneath the caudal vertebrae would be the caudal ventral osteoderms and those beneath the ‘dorsal’ vertebrae would be the dorsal ventral osteoderms. This is nonsensical so I suggest a new term be used for what have been known as the dorsal vertebrae and osteoderms in aetosaurians. The terms “thoracic” and “lumbar” vertebrae reflect the chest and loin areas respectively and are assigned depending on the presence or absence of free ribs. This is not readily applicable to pseudosuchians which have ribs through the entire series. Instead the term trunk vertebrae is used, which is commonly used for amphibians and lepidosaurs, which also tend to have a ribs throughout the entire series (e.g., Wake, 1992). The osteoderms above the trunk vertebrae are the dorsal trunk paramedian and dorsal trunk lateral osteoderms. The osteoderms located beneath the trunk vertebrae are the ventral trunk osteoderms and consists of numerous columns of osteoderms (Fig. 3; Walker, 1961). Heckert et al. (2010) utilized the term ventral thoracic osteoderms, which effectively solves the ‘ventral dorsal’ problem; however, the term ventral trunk osteoderms is preferred here to maintain consistency with the term dorsal trunk osteoderms.

The osteoderms above the caudal vertebrae are termed the dorsal caudal osteoderms and consist of paramedian and lateral columns (Fig. 3; Long & Ballew, 1985). The osteoderms beneath the caudal vertebrae are the ventral caudal osteoderms (Heckert et al., 2010) and also consist of paramedian and lateral columns behind the cloacal area (fourth row) to the tip of the tail (Jepson, 1948; Walker, 1961), the first two lateral rows bear spines in Typothorax coccinarum (Heckert et al., 2010). An assemblage of irregular shaped osteoderms located anterior to the cloacal area is preserved in Stagonolepis robertsoni, Aetosaurus ferratus, and Typothorax coccinarum (Walker, 1961; Schoch, 2007; Heckert et al., 2010), which can be called the cloacal osteoderms. Small masses of irregular shaped osteoderms cover the limb elements of aetosaurians (e.g., Heckert & Lucas, 1999; Schoch, 2007; Heckert et al., 2010). These have collectively been termed as simply appendicular osteoderms. However, when found in articulation they can be differentiated by the limb that is covered, including the humeral, radioulnar, femoral, and tibiofibular osteoderms (Hill, 2010).

Systematic paleontology

Archosauria Cope, 1869 sensu Gauthier & Padian, 1985.

Pseudosuchia Zittel 1887–1890 sensu Gauthier & Padian, 1985.

Aetosauria Marsh, 1884 sensu Parker, 2007.

Stagonolepididae Lydekker, 1887 sensu Heckert & Lucas, 2000.

Scutarx deltatylus: Parker, 2016

(Figs. 4–29)

1985 Calyptosuchus wellesi: Long & Ballew, p. 54, Figs. 13A and 15, Pl. 5.

1995 Stagonolepis wellesi: Long & Murry, p. 82, Figs. 71B, 72B and 72E.

2005 Stagonolepis wellesi: Parker & Irmis, p. 49, Fig. 4A.

2005a Stagonolepis wellesi: Parker, p. 44.

2005b Stagonolepis wellesi: Parker, p. 35.

2006 Stagonolepis wellesi: Parker, p. 53.

2011 Calyptosuchus wellesi: Parker & Martz, p. 242.

2013 Calyptosuchus wellesi: Martz et al., p. 342, Figs. 7A–7D.

2014 Calyptosuchus wellesi: Roberto-Da-Silva et al., p. 247.

2016 Scutarx deltatylus: Parker, p. 27, Figs. 2–5.

Holotype: PEFO 34616, articulated posterior portion of a skull with the braincase; detached left nasal; cervical and dorsal trunk paramedian and dorsal trunk lateral osteoderms; ventral osteoderms, rib fragments, and paired gastral ribs.

Paratypes: PEFO 31217, much of a postcranial skeleton including vertebrae, ribs, pectoral and pelvic girdles, osteoderms; PEFO 34919, much of a postcranial skeleton including vertebrae, ribs, osteoderms, girdle fragments, ilium; PEFO 34045, much of a postcranial skeleton including vertebrae, ribs, and osteoderms.

Referred specimens: UCMP 126844, 10 partial dorsal trunk paramedian osteoderms (PFV 162, Lot’s Wife beds, Sonsela Member, Chinle Formation, PEFO, Arizona); UCMP 84916, partial left dorsal trunk paramedian osteoderm (PFV 146, lower part of the Sonsela Member, Chinle Formation, Billings Gap, Arizona; UCMP 36656, UCMP 35738, dorsal trunk paramedian and dorsal trunk lateral osteoderms (UCMP locality V3205; lower part of the Sonsela Member, Chinle Formation, 11 km north of Nazlini, Arizona); TTU P-09240, left and right dorsal trunk paramedian osteoderms (Post Quarry, Cooper Canyon Formation, Dockum Group, Texas).

Type locality, Horizon, and Age: PFV 255 (The Sandcastle), PEFO, Arizona; lower part of the Sonsela Member, Chinle Formation; Adamanian biozone, Norian, ∼217 Ma (Ramezani et al., 2011).

Diagnosis: From Parker (2016): Medium-sized aetosaurian diagnosed by the following autapomorphies: the cervical and dorsal trunk paramedian osteoderms bear a strongly raised, triangular tuberosity in the posteromedial corner of the dorsal surface of the osteoderm; the occipital condyle lacks a distinct neck because the condylar stalk is mediolaterally broad; the frontals and parietals are very thick dorsoventrally; and there is a distinct fossa or recess on the lateral surface of the ilium between the supraacetabular crest and the posterior portion of the iliac blade. An additional possible autapomorphy is that the base of the cultriform process of the parabasisphenoid bears deep lateral fossae; however, in more complete, articulated aetosaur skulls (e.g., SMNS 19003) the parabasisphenoid is articulated with the pterygoids making this difficult to observe. Scutarx deltatylus can also be differentiated from other aetosaurs a unique combination of characters including moderately wide (W:L ∼2.7/1) dorsal trunk paramedian osteoderms with a strongly raised anterior bar that possesses anteromedial and anterolateral processes (shared with all aetosaurians except Desmatosuchini); osteoderm surface ornamentation of radiating ridges and pits that emanate from a posterior margin contacting a dorsal eminence (shared with Calyptosuchus wellesi, Stagonolepis robertsoni, Adamanasuchus eisenhardtae, Neoaetosauroides engaeus, and Aetosauroides scagliai); lateral trunk osteoderms with an obtuse angle between the dorsal and lateral flanges (shared with non-desmatosuchines); a dorsoventrally short pubic apron with distinct obturator and thyroid fenestrae (shared with Stagonolepis robertsoni); and an extremely anteroposteriorly short parabasisphenoid, with basal tubera and basipterygoid processes almost in contact and a reduced cultriform process (shared with Desmatosuchus).

Description

Skull

Much of the posterodorsal portion of the skull is present in PEFO 34616 (Figs. 4–10). Elements preserved include much of the left nasal, both frontals (the right is incomplete), both postfrontals, the left parietal (badly damaged), the left and right squamosals, the right postorbital, a portion of the left postorbital, and a nearly complete occipital region and braincase. The skull was already heavily eroded when discovered and although the skull roof/braincase portion was collected in situ, the remaining elements had to be carefully pieced together from many fragments collected as float. Accordingly many of the skull roof elements are incomplete.

Figure 4 Photos and interpretive sketches of the left nasal (PEFO 34616) in dorsal (A) and ventral (B) views.

Arrows point anteriorly and scale bars equals 1 cm. Abbreviations: en, external nares; fr, frontal; la, lacrimal; mx, maxilla; s., suture with listed element.

Figure 5 Photo and interpretive sketch of posterodorsal portion of the skull of Scutarx deltatylus (PEFO 34616) in dorsal view.

Scale bar equals 1 cm. Abbreviations: bo, basioccipital; gr, groove; ex, exoccipital; lfr, left frontal; ls, laterosphenoid; na, nasal; orb, orbit; pa, parietal; par, paroccipital process of the opisthotic; plpr, palpebral; po, postorbital; pof, postfrontal; pr, prootic; prf, prefrontal; rfr, right frontal; s., suture with listed element; so, supraoccipital; sq, squamosal; stf; supratemporal fenestra.

Figure 6 Partial skull of Scutarx deltatylus (PEFO 34616) in right lateral view.

Scale bar equals 1 cm. Abbreviations: bo, basioccipital; bpt, basipterygoid processes; bsr, basisphenoid recess; bt, basal tubera; cp, cultriform process; fr, frontal; ls, laterosphenoid; na, nasal; of, orbital fossa; orb, orbit; pa, parital; palp, palpebral; po, postorbital; pof, postfrontal; pr, prootic; prf, prefrontal; qj, quadratojugal; qu, quadrate; sq, squamosal; stf, supratemporal fenestra; uc, unossified cleft of the basal tubera.

Figure 7 Partial skull of Scutarx deltatylus (PEFO 34616) in ventral view.

Scale bar equals 1 cm. Abbreviations: bo, basioccipital; btp, basipterygoid processes; bsr, basisphenoid recess; bt, basal tubera; cp, cultriform process; f., fossa for specified element; lfr, left frontal; ls, laterosphenoid; mf, metotic fissure; na, nasal; of, orbital fossa; orb, orbit; pa, parietal; palp, palpebral; par, paroccipital process of the opisthotic; po, postorbital; pof, postfrontal; pr, prootic; prf, prefrontal; qj, quadratojugal; qu, quadrate; rfr, right frontal; sq, squamosal; stf, supratemporal fenestra; uc, unossified cleft of the basal tubera.

Figure 8 Partial skull of Scutarx deltatylus (PEFO 34616) in posterior view.

Scale bar equals 1 cm. Abbreviations: bo, basioccipital; bpt, basipterygoid processes; bs, basisphenoid; ex, exoccipital; ex.pr; exoccipital prong; fm, foramen magnum; lfr, left frontal; pa, parietal; par.op, paroccipital process of the opisthotic; po, postorbital; rfr, right frontal; sq, squamosal.

Figure 9 Braincase of Scutarx deltatylus (PEFO 34616) in ventrolateral view.

Scale bar equals 1 cm. Abbreviations: bpt, basipterygoid processes; bsr, basisphenoid recess; bt, basal tubera; cc, cotylar crest; clp, clinoid process; cp, cultriform process; crp, crista prootica; fo, foramen ovale; hypf, hypophyseal fossa; ic, exit area of the internal carotid artery; lfr, left frontal; lr, lateral ridge; ls, laterosphenoid; mf, metotic foramen; na, nasal; oc, occipital condyle; orb, orbit; pa, parietal; par, paroccipital process of the opisthotic; po, postorbital; pr, prootic; prf, prefrontal; psr, parasphenoid recess; rfr, right frontal; s., suture with designated element; sq, squamosal; uc, unossified cleft of the basal tubera; V, passageway for the Trigeminal nerve.

Much of the skull appears to have separated originally along some of the sutures, notably those between the prefrontal-frontal, squamosal-quadrate, and postorbital-quadratojugal contacts. The left frontoparietal suture is also visible because of bone separation, and the sockets in the squamosals for reception of the proximal heads of the quadrates are well-preserved. Thus, the skull appears to have mostly fallen apart before burial and many of the anterior and ventral elements were presumably scattered and lost during disarticulation, with the exception of the left nasal, which is represented as an isolated piece. The skull of Scutarx deltatylus includes a well-preserved braincase, which is described in detail below. Sutures are difficult to observe because of the state of preservation of the specimen, and the skull of Longosuchus meadei (TMM 31185-98) was used to infer the locations of various sutures, based on observable landmarks present in PEFO 34616.

Nasal

The proximal half of the left nasal is preserved, consisting of the main body and the posterior portion of the anterior projection through the mid-point of the external naris (Fig. 4). The main body is dorsoventrally thick and the entire element is slightly twisted dorsomedially so that the dorsal surface is noticeably concave. Any surface ornamentation is obscured by a thin coating of hematite. The midline symphysis is straight and slightly rugose (Fig. 4). The lateral surface is damaged along the lacrimal suture; however, more anteriorly, the sutural surface for the ascending process of the maxilla is preserved and is strongly posteroventrally concave (Fig. 4). Anteriorly the nasal narrows mediolaterally where it forms the dorsal margin of the external naris. The ventral process of the nasal that borders the posterior edge of the naris is missing its tip but it is clear from what is preserved that it was not elongate as in Aetosauroides scagliai (PVL 2059), but rather short as in Stagonolepis olenkae (ZPAL AbIII/346).

Frontal

Both frontals are present, with the left nearly complete and the right missing the posterior portion (Fig. 5). The extreme dorsoventral thickness of the element is evident, as the dorsoventral thickness is 0.35 times the midline length of the element. The frontals appear to be hollow; however, this is most likely from damage during deposition and subsequent weathering before the skull roof was collected and pieced back together. In dorsal view the posterior margin of the frontal is slanted posterolaterally as in Stagonolepis robertsoni (Walker, 1961) so that the lateral margin of the frontal is longer than the medial margin, forming a distinct posterolateral process (Fig. 5). The anterior portion of that process meets the postfrontal laterally and the parietal posteriorly as in Stagonolepis olenkae (Sulej, 2010). Just anterior to the posterolateral process the frontal forms the dorsal margin of the orbit. The position of the suture with the postfrontal is not clear, but it should have been present as in all other aetosaurians.

The dorsal surfaces of the frontals are rugose, ornamented with deep pits, some associated with more elongate grooves. Laterally above the round orbits and anteriorly there are wider, anteroposteriorly oriented grooves as in Stagonolepis olenkae (Sulej, 2010). These grooves demarcate a raised central portion of the frontals as described for Stagonolepis robertsoni by Walker (1961). The anterolateral margins of the frontals are dorsoventrally thick, rugose, anteromedially sloping areas that are bounded posteriorly by a thin curved ridge. These are the sutures for the prefrontals (Figs. 5 and 6). There is no clear evidence for articulation of a palpebral bone at this position as in Stenomyti huangae (Small & Martz, 2013), but the posterior-most portion of the articular surface (Fig. 6) is probably a suture for a palpebral as in Longosuchus meadei (TMM 31184-98). The anterior margins of the frontals are thick and rugose for articulation with the nasals (Figs. 5 and 7). The frontal/nasal suture is nearly transverse. The frontal also lacks the distinct, raised midline ridge present in Stenomyti huangae (Small & Martz, 2013).

The ventral surfaces of the frontals are broadly ventrally concave and smooth (Fig. 7). Medial to the orbital fossa is a distinct, slightly curved ridge that is the articulation point with the laterosphenoid.

Postfrontal

The postfrontals are small, roughly triangular bones that form the posterodorsal margin of the orbit. Both are certainly preserved in PEFO 34616, as in all aetosaurians, but the positions of their sutures cannot be determined.

Parietal

The dorsal portions of both parietals are mostly missing, although the posterolateral corner of the left one remains as well as a small fragment of the posterior portion of the right where it contacts the dorsal process of the squamosal (Fig. 5). The frontal/parietal suture is visible along the posterior margin of the frontals, so it is clear that these elements were not fused. The posterolateral portion forms the dorsal border of the supratemporal fenestra, but few other details are visible.

The posterior flanges of both parietals are preserved (Fig. 8). Their posteroventrally sloping surfaces form the upper portion of the back of the skull. Ventrally, they contact the paroccipital processes of the opisthotics. There is no evidence for the postemporal fenestrae, which may have been obliterated by slight ventral crushing of the skull roof. The parietal flanges contact the supraoccipital medially and the posterior process of the squamosal laterally. The upper margins are damaged so that the presence of a shelf for articulation of the nuchal paramedian osteoderms cannot be confirmed.

Squamosal

The majority of both squamosals is present. As is typical for aetosaurians the squamosals are elongate bones that are fully exposed in lateral view, forming the posterior corner of the skull, as well as the posteroventral margin of the supratemporal fenestra (Fig. 6). The anterior and posterior portions are separated by a dorsoventrally thin neck. The anterior portion divides into two distinct rami, a large, but mediolaterally thin, ventral lobe that presumably contacted the upper margin of the quadratojugal, and a much smaller triangular dorsal ramus that forms much of the anteroventral margin of the supratemporal fenestra. These two rami are separated by a posterior process of the postorbital. On the right side of PEFO 34616, the dorsal ramus is broken, clearly showing the articulation with the postorbital and exposing the prootic in this view (Fig. 6). The ventral margin of the main body is concave and bears a flat surface that is the articulation surface with the quadrate (s.qu; Fig. 7). Anterior to that articular surface the ventral margin of the anterior portion of the squamosal is confluent with the ventral margin of the postorbital. This arrangement suggests that the squamosal contributed little if anything to the margin of the infratemporal fenestra. This is similar to the condition in Stagonolepis robertsoni (Walker, 1961) and differs from that in Stenomyti huangae (Small & Martz, 2013) in which the ventral margin of the squamosal is situated much lower that the ventral margin of the postorbital, and the squamosal contributes significantly to the margin of the infratemporal fenestra.

The posterior portion of the squamosal expands posterodorsally into dorsal and ventral posterior processes. The dorsal process forms the posterior border of the supratemporal fenestra and is mediolaterally thickened with a smooth anterior concave area that comprises the supratemporal fossa. The apex of the upper process contacts the parietal. The ventral posterior process forms a small hooked knob that projects off of the back of the skull. Medial to this is a deep pocket in the medial surface of the squamosal that receives the dorsal head of the quadrate. Dorsomedial to this pocket is the contact between the squamosal and the distal end of the paroccipital process of the opisthotic (Figs. 5 and 7).

Postorbital

A portion of the left and almost the complete right postorbital are preserved in PEFO 34616 (Figs. 5–7). They are mediolaterally thin, triradiate bones that contact the postfrontal and parietal dorsally, the jugal anteriorly, and the squamosal posteriorly. The upper bar forms the posterior margin of the orbit and the anterior margin of the supratemporal fenestra. The posterior process is triangular and inserts into a slot in the anterior portion of the squamosal. The ventral margin is flat, and forms the dorsal border of the infratemporal fenestra and more anteriorly that edge bears an articular surface with the jugal. The tip of the anterior process is broken, but it would have overlain the posterior process of the jugal and formed the posteroventral margin of the orbit. The postorbital of Scutarx appears to lack to broad ventral contact with the quadratojugal (Fig. 6) as in Paratypothorax (SMNS 19003) and Aetosaurus (Schoch, 2007), and instead was like Desmatosuchus spurensis (Small, 2002) and Stagonolepis (Walker, 1961; Sulej, 2010) where an anterior process of the squamosal separates the postorbital from the quadratojugal.

Supraoccipital

The supraoccipital is present but poorly preserved (Fig. 8). A median element, it forms much of the dorsal portion of the occiput, but appears to be at least partially excluded from the roof of the foramen magnum. Laterally it contacts the parietal flanges and ventrally the otooccipitals.

Exoccipital/opisthotic

The exoccipitals and opisthotics are fused into a single structure, the otooccipital. The exoccipital portions form the lateral margins of the foramen magnum (Fig. 8). A protuberance, or prong, is present on the left exoccipital at the dorsolateral corner of the foramen magnum (Figs. 5 and 8). The presence of similar structures in Neoaetosauroides engaeus (e.g., PVL 5698) was noted by Desojo & Báez (2007), and interpreted by them to be facets for reception of the proatlantes. Those authors considered the facets located on the supraoccipital; however, in Longosuchus meadei (TMM 31185-84) they are located on the exoccipital and the same appears to be true for PEFO 34616.

Anteriorly, a strong lateral ridge forms the posteroventral margin of the ‘stapedial groove’ as is typical for aetosaurs (Gower & Walker, 2002). In aetosaurians there are typically two openings for the hypoglossal nerve (XII) that straddle the lateral ridge (Gower & Walker, 2002); however, they are not apparent in PEFO 34616, and where the posterior opening of the left side should be situated there is a fragment of bone missing.

Both paroccipital processes are present and well-preserved (Figs. 5–8). They are mediolaterally short (14 mm) and stout, dorsoventrally taller than anteroposteriorly long (8 mm tall, 4 mm long), and contact the parietal flanges dorsally and the squamosal laterally. The distal end expands slightly dorsoventrally (Fig. 8). The posterior surface is flat and distally the process forms the posterior border of the pocket for reception of the quadrate head, therefore there was a sizeable contact between the opisthotic and the quadrate.

The proximoventral portion of the paroccipital process opens into the ‘stapedial groove.’ That groove continues into the main body of the opisthotic, bounded by the lateral ridge of the exoccipital posteroventrally and the crista prootica anterodorsally (Fig. 9). Here there is a large opening for the fenestra ovalis and the metotic foramen; however, the two cannot be distinguished because the ventral ramus of the opisthotic that divides the two openings in aetosaurians (Gower & Walker, 2002) is not preserved (Fig. 9). It is not clear if the ventral ramus was never originally preserved or if it was removed during preparation of the braincase. Thus, the perilymphatic foramen is not preserved as well. The embryonic metotic fissure is undivided in aetosaurs and therefore the glossopharyngeal, vagal, and accessory (IX, X, XI) nerves and the jugular vein would have exited the braincase via a single opening, the metotic foramen (Gower & Walker, 2002; also see Rieppel, 1985; Walker, 1990). Just lateral to the metotic foramen on the ventral surface of the crista prootica there should be a small opening for the facial nerve (VII); however, it is not visible through the hematite build-up on the lateral wall of the cranium.

A second distinct groove extends from the ventral border of the fenestra ovalis anteroventrally along the lateral face of the parabasisphenoid to the posterodorsal margin of the basipterygoid process, and is bordered anterodorsally by the anteroventral continuation of the crista prootica (Fig. 9). The termination of that groove houses the entrance of the cerebral branch of the internal carotid artery (Gower & Walker, 2002; Sulej, 2010).

Prootic

The entire braincase is slightly crushed and rotated dorsolaterally so that the left side of the otic capsule is easier to view (Fig. 9). Both prootics are preserved. Posteriorly, the prootic overlaps the opisthotic medially, and ventrolaterally forms a thin ridge (crista prootica), which is bounded ventrally by the upper part of the ‘stapedial groove’ and the groove in the parabasisphenoid leading to an opening for the internal carotid. Anteroventrally, the prootic meets the anterior portion of the parabasisphenoid, just posterior to the hypophyseal fossa. Anteriorly and anterodorsally, the prootic meets the laterosphenoid and dorsally it is bounded by the parietal. The uppermost margin is deformed by a thick anteroposteriorly oriented mass of bone, which could represent crushing of the parietal margin. Just posterior to the anterior suture with the laterosphenoid is the opening for the trigeminal nerve (V) which is deformed and closed by crushing (Fig. 9). In PEFO 34616 the opening for the trigeminal nerve is completely enclosed by the prootic. This is similar to the condition in Stagonolepis olenkae (Sulej, 2010), Stagonolepis robertsoni (Walker, 1961), and Longosuchus meadei (TMM 31185-98) and appears to be typical for all aetosaurs; however, Small (2002) shows the trigeminal opening subdivided in the skull of Desmatosuchus smalli although he does not describe it.

Laterosphenoid

The laterosphenoids are ossified but poorly preserved. On the left side anterodorsal to the opening for the trigeminal nerve (V), there is the cotylar crest, which is crescentic and opens posteriorly (Fig. 9). No other details of the laterosphenoid can be determined.

Basioccipital/parabasisphenoid

The basioccipital and parabasisphenoid are complete and together comprise the best preserved and most distinctive portion of the braincase in Scutarx deltatylus (Fig. 10). The occipital condyle is transversely ovate in posterior view rather than round like in other aetosaurs such as Longosuchus meadei (TMM 31185-98) and Desmatosuchus smalli (TTU P-9024). The dorsal surface is broad with a wide shallow groove for the spinal cord.

Figure 10 Parabasisphenoid of Scutarx deltatylus (PEFO 34616) in ventral view.

Scale bar equals 1 cm. Abbreviations: bpt, basipterygoid processes; bsr, basisphenoid recess; bt, basal tubera; cp, cultriform process; crp, crista prootica; f., fossa for specified element; lfr, left frontal; lr, lateral ridge; ls, laterosphenoid; of, orbital fossa; orb, orbit; par, paroccipital process of the opisthotic; po, postorbital; prf, prefrontal; pr, prootic; prf, prefrontal; psr, parasphenoid recess; quadrate; rfr, right frontal; sq, squamosal; ssr, subsellar recess; stf, supratemporal fenestra; uc, unossified cleft of the basal tubera.

The condylar stalk is also broad (25 mm wide), and wider than the condyle. Thus there is no distinct ‘neck,’ nor does a sharp ridge delineate the condyle from the stalk as in Longosuchus meadei (TMM 31185-98; Parrish, 1994) or Desmatosuchus smalli (TTU P-9024; Small, 2002). The ventral surface of the condylar stalk bears two low rounded ‘keels’ separated by a shallow, but distinct, oblong pit. The broad stalk, lack of a distinct neck, and ventral keels all appear to be autapomorphic for Scutarx deltatylus. Anterolaterally the condylar stalk expands laterally to form the ventral margin of the metotic fissure. The contacts with the exoccipitals are dorsal and posterior to that margin.

The right basal tuber of the basioccipital is present, but the left is missing. The basioccipital tuber is separated from the crescentic basal tuber of the parabasisphenoid by an unossified cleft, typical for aetosaurians and other suchians (Fig. 10; Gower & Walker, 2002). The basal tubera of the basioccipital are divided medially by an anteroposteriorly oriented bony ridge that bifurcates anteriorly to form the crescentic basal tubera of the parabasisphenoid and enclose the posterior portion of the basisphenoid recess (sensu Witmer, 1997). Posteriorly that bony ridge is confluent with the posteriorly concave posterior margin of the basioccipital basal tubera (Fig. 10). The short, anterolaterally directed basipterygoid processes are located anteriorly and in contact posteriorly with the anterior margin of the basal tubera of the parabasisphenoid. The upper portion of the distal end of the left basipterygoid process is broken, but the right is complete and bears a slightly expanded and slightly concave distal facet that faces anterolaterally to contact the posterior process of the pterygoid.

The basipterygoid processes and the basal tubera are positioned in the same horizontal plane (Fig. 9), which is typical for aetosaurians and differs significantly from the condition in Revueltosaurus callenderi (PEFO 34561) and Postosuchus kirkpatrickorum (TTU P-9000; Weinbaum, 2011; emend Parker, 2016) in which the basicranium is oriented more vertically, with the basipterygoid processes situated much lower dorsoventrally than the basal tubera.

Scutarx deltatylus differs from aetosaurians such as Stagonolepis robertsoni (MCZD 2), Neoaetosauroides engaeus (PVL 5698), and Aetosauroides scagliai (PVSJ 326) in that there is a broad contact between the basal tubera and the basipterygoid processes and that the basipterygoid processes are not elongate (Fig. 10). This is nearly identical to the condition in Desmatosuchus smalli (TTU P-9023) and Desmatosuchus spurensis (UMMP 7476; Case, 1922). There are two basicrania (UCMP 27414, UCMP 27419) from the Placerias Quarry with widely separated (anteroposteriorly) basal tubera and (elongate) basipterygoid processes that apparently do not pertain to either Desmatosuchus or Scutarx deltatylus, and may belong to Calyptosuchus wellesi. This would demonstrate a potential important braincase difference between Calyptosuchus wellesi and Scutarx deltatylus, despite the nearly identical structure of the osteoderms shared between these two taxa.

In the anteroposteriorly short area between the basal tubera and the basipterygoid processes, a deep, subrounded fossa (Fig. 10) comprises the basisphenoid recess (= median pharyngeal recess of Gower & Walker (2002); = parabasisphenoid recess of Nesbitt (2011)), which is formed by the median pharyngeal system (Witmer, 1997). The presence of a ‘deep hemispherical fontanelle’ (= basisphenoid recess) between the basal tubera and the basipterygoid processes has been proposed as a synapomorphy of Desmatosuchus and Longosuchus (Parrish, 1994), but, as discussed by Gower & Walker (2002), that condition is present in many archosauriforms. The number of aetosaurian taxa with this feature was expanded by Heckert & Lucas (1999), who also reported that a ‘hemispherical fontanelle’ is absent in Typothorax and Aetosaurus. Unfortunately they did not list catalog numbers for examined specimens, and scoring of character occurrences cannot be replicated. The basisphenoid recess is actually present in Aetosaurus (Schoch, 2007), Paratypothorax (SMNS 19003), Neoaetosauroides (PVL 5698), and Typothorax (TTU P-9214; Martz, 2002). Thus, the presence of that recess is an aetosaurian synapomorphy.

Small (2002) found the shape and size of the basisphenoid recess to be variable in his hypodigm of Desmatosuchus haplocerus, and recommended that the character be dropped from phylogenetic analysis pending further review. However, rather than utilizing the presence or absence of the structure, it has been proposed that the shape and depth may be of phylogenetic significance (Gower & Walker, 2002). As noted above, it appears that there are two types of aetosaurian basicrania, those with anteroposteriorly short parabasisphenoids and those with long parabasisphenoids. These differences were used as rationale for splitting Desmatosuchus haplocerus into two species (Parker, 2005b). Among taxa with short parabasisphenoids, Scutarx deltatylus (PEFO 34616) and Desmatosuchus spurensis (UMMP 7476) have deep, more or less round basisphenoid recesses, and Desmatosuchus smalli has a shallow subtriangular recess. In Longosuchus meadei (TMM 31185-98) the recess is round and shallow. Among taxa with elongate basisphenoids, Aetosauroides scagliai (PVSJ 326) has a shallow, round recess and Tecovasuchus chatterjeei (TTU P-545) has a deep, round recess. However, in Coahomasuchus kahleorum (NMMNH P-18496; TMM 31100-437), which has an elongate basisphenoid, the recess has the form of a moderately deep, anteroposteriorly elongate oval (Desojo & Heckert, 2004; pers. obs. of TMM 31100-437). Thus, the shape of this structure is highly variable and most likely not phylogenetically informative, although the elongate form of the recess in C. kahleorum may prove autapomorphic.

Anterior to the basisphenoid recess and between the bases of the basipterygoid processes there is another shallow, anteroventrally opening recess (Fig. 10). This recess is at the base of the parasphenoid process, in the same position as the subsellar recess in theropod dinosaurs (Witmer, 1997; Rauhut, 2004) and may be homologous to the latter. However, the function and origin of the recess are not understood (Witmer, 1997). It is also present in Neoaetosauroides engaeus (PVL 5698) and may have a broader distribution within Aetosauria.

Dorsal to the basipterygoid processes, two crescentic and dorsally expanding clinoid processes flank the circular, concave hypophyseal fossa, which housed the pituitary gland (Fig. 9). No openings are visible because of poor preservation, but the dorsum sellae should be pierced by two canals for the abducens (VI) nerves (Hopson, 1979; Gower & Walker, 2002). At the base of the hypophyseal fossa in Stagonolepis robertsoni (MCZD 2) and Longosuchus meadei (TMM 31185-98) there is a triangular flange of bone termed the parabasisphenoid prow (Gower & Walker, 2002). This structure is mostly eroded in PEFO 34616, although its base is preserved as a small dorsal protuberance.

Anterior to this, the cultriform process of the parasphenoid is completely preserved (Figs. 9 and 10). This structure is delicate and usually missing, obscured, or in articulation with the pterygoids in the few known aetosaur skulls, making comparisons difficult. However, the process is notably short in PEFO 34616, barely extending past the anterior margins of the orbits (Fig. 9). In PEFO 34616 the basisphenoid has a length of 34.2 mm, whereas the cultriform process measures 20.2 mm in length (cultriform process/basisphenoid ratio = 0.59). This is noticeably different from the parabasisphenoid in Aetosauroides scagliai (PVSJ 326) which has a basisphenoid length of 51 mm and a cultriform process length of at least 63 mm, although the anterior end of the process is concealed (ratio = 1.23) beneath the left pterygoid. The cultriform process is also preserved in Desmatosuchus spurensis (UMMP 7476), which has a relatively short parabasisphenoid and a cultriform process/basisphenoid ratio of 0.96.

The cultriform process is elongate and tapers anteriorly. It is Y-shaped in cross-section with a ventral ridge, and dorsal trough for the ethmoid cartilage. Its posterolateral margins bear distinct oval recesses bound posterodorsally by strong ridges that are confluent with the posterodorsal edge of the process (Figs. 9 and 10). Thus, the process is broader posteriorly, with these recesses contributing greatly to the thinning of the element anteriorly. The parasphenoid recesses appear to be unique to PEFO 34616, although the general lack of known aetosaurian cultriform processes, or their preservation articulated with the pterygoids, makes it difficult to determine this with certainty.

Postcranial skeleton

Vertebrae

Cervical series

Post-axial cervicals

Two articulated cervical vertebrae are preserved in PEFO 31217 (Fig. 11). Although both are crushed mediolaterally, they are nearly complete and preserve many details. The centra are taller than long (Fig. 11A) suggesting they represent part of the anterior (post-axial) series (i.e., positions 3–6). Most notably, the difference in dimensions is not as pronounced as in Typothorax coccinarum and Neoaetosauroides engaeus, in which the centra are greatly reduced in length (Long & Murry, 1995; Desojo & Báez, 2005; Heckert et al., 2010). The centrum faces are subcircular in anterior and posterior views and slightly concave, with slightly flared rims (Figs. 11B and 11C). The ventral surface of each centrum consists of two concave, ventromedially inclined, rectangular surfaces divided by a sharp and deep mid-line keel (Fig. 11D).

Figure 11 Articulated anterior post-axial vertebrae of Scutarx deltatylus (PEFO 31217) in posterolateral (A), posterior (B), anterior (C), and ventral (D) views.

Scale bar equals 1 cm. Abbreviations: diap, diapophysis; k, keel; nc, neural canal; ns, neural spine; parp, parapophysis; pocdf, postzygapophyseal centrodiapophyseal fossa; posz, postzygapophysis; prez, prezygapophysis; spof, spinopostzygapophyseal fossa; spol, spinopostzygapophyseal lamina; tpol, intrapostzygapophyseal lamina.

The short parapophyses are oval in cross-section and situated at the anteroventral corners of the centrum. The parapophyses are directed posteriorly, and each forms the beginning of a prominent ridge that extends to the posterior margin of the centrum. The lateral faces of the centra are concave mediolaterally and dorsoventrally form discrete, but shallow, lateral fossae that contact the neural arch dorsally (Fig. 11A). However, PEFO 31217 lacks the deep lateral fossae, which are considered an autapomorphy of Aetosauroides scagliai (Desojo & Ezcurra, 2011). The neurocentral sutures are not apparent on this specimen, suggesting closure of the sutures and that this individual is osteologically ‘mature’ although this cannot be completely confirmed without histological sectioning of the sutural contact (Brochu, 1996; Irmis, 2007).

The diapophyses are centrally located at the base of the neural arch (Fig. 11B). The best preserved vertebra shows that they are slightly elongate, oval in cross-section, and curved ventrolaterally. Because none of the diapophyses appears to be complete their exact length cannot be determined. The neural canal is round in posterior view (Fig. 11C) rather than rectangular as in Desmatosuchus spurensis (UMMP 7504). The entire neural arch is taller than the corresponding centrum face. The zygapophyses are well-formed, elongate, and oriented at approximately 45° from the horizontal.

Aetosaurian vertebrae bear several vertebral laminae and associated fossae. The terminology for these structures follows Wilson (1999) and Wilson et al. (2011). There is a weakly developed posterior centrodiapophyseal lamina (pcdl) that originates at the posteroventral corner of the diapophysis and continues posteroventrally to the posterior edge of the neurocentral suture. The only other apparent vertebral laminae are paired intrapostzygapophyseal laminae (tpol) that originate on the posteroventral surface of the postzygapophyses and form two sharp ridges (laminae) that meet at the dorsomedial margin of the neural canal (Fig. 11B). Those laminae delineate the medial margins of a pair of distinct lateral fossae, called the postzygapophyseal centrodiapophyseal fossae (pocdf), as well as a sizeable medial fossa, called the spinopostzygapophseal fossa (spof). This represents the first recognition of distinct intrapostzygapophyseal laminae in an aetosaurian. Desmatosuchus spurensis (MNA V9300) has struts of bone from the dorsomedial margins of the postzygapophyses that join medially and then extend ventrally as a single thickened unit to form a Y-shaped hyposphene (Parker, 2008a: Fig. 10A), similar to the pattern formed by the intrapostzygapophyseal laminae in Scutarx deltatylus. Thus, it is possible that the structure of the hyposphene in aetosaurians is homologous (i.e., the hyposphene is actually formed by paired vertebral laminae) with the presence of paired (but not joined) intrapostzygapophyseal laminae, but this interpretation requires further investigation.

The neural spines are not complete; however, the base of the one on the second preserved vertebra shows that the spine was anteroposteriorly elongate, with prominent spinopostzygapophyseal laminae (spol) that are confluent with the dorsal surfaces of the postzygapophyses (Fig. 11B). Spinopostzygapophyseal laminae are also present on the cervical vertebrae of Desmatosuchus spurensis (Parker, 2008a).

Trunk series

Mid-trunk vertebrae

Four mid-trunk vertebrae are preserved in PEFO 34045. In aetosaurs the cervical to trunk transition occurs when the parapophysis fully migrates from the base of the neural arch, laterally onto the ventral surface of the transverse process (Case, 1922; Parker, 2008a). PEFO 34045/FF-51 is well preserved, missing only the postzygapophyses (Figs. 12A–12C). The articular faces of the centra are round and slightly concave with broad flaring rims. The centrum is longer (45.78 mm) than tall (41.81 mm), its lateral faces are deeply concave, and its ventral surface is narrow and smooth. The neural canal is large and in anterior view, the margins of the neural arch lateral to the canal are mediolaterally thin with sharp anterior edges.

Figure 12 Trunk vertebrae of Scutarx deltatylus.

(A–C) PEFO 34045/FF-51, mid-trunk vertebra in posterior (A), anterior (B), and lateral (C) views. Scale bar equals 1 cm. Abbreviations: b., broken designated element; cpof, centropostzygapophyseal fossa,; cprf, centroprezygapophyseal fossa; diap, diapophysis; nst, neural spine table; parp, parapophysis; podl, postzygadiapophyseal lamina; posz, postzygapophysis; prez, prezygapophysis; pro, projection; sprf, spinoprezygapophyseal fossa; spol, spinopostzygapophyseal lamina.

The prezygapophyses are inclined at about 45° from the horizontal and are confluent laterally with a short horizontally oriented prezygadiapophyseal lamina (prdl) that terminates laterally at the parapophysis (Fig. 12B). Between the prezygapophyses and ventral to the base of the neural spine there is a well-developed broad, sub-triangular spinoprezygapophyseal fossa (sprf). In combination with the flat prezygapophyses this creates a broad shelf for reception of the posterior portion of the neural arch of the preceding vertebra (Fig. 12B). There is a horizontal, ventral bar that roofs the opening of the neural canal between the ventromedial edges of the prezygapophyses (Fig. 13A); thus, there is no developed hypantrum as in Desmatosuchus spurensis or Aetobarbakinoides brasiliensis (Desojo, Ezcurra & Kischlat, 2012; Parker, 2008a). The ventral bar also occurs in Stagonolepis robertsoni (Walker, 1961: Fig. 7J). Ventrolateral to the prezygapophysis there is a deep centroprezygapophyseal fossa (cprf), which is bordered posteriorly by the main strut of the transverse process (Fig. 12B). Although the positions of these fossae appear homologous with those of saurischian dinosaurs because they share distinct topological landmarks, it is not clear if these features are similarly related to the respiratory system as they are in saurischians (Butler, Barrett & Gower, 2012; Wilson et al., 2011).

Figure 13 Trunk vertebrae of Scutarx deltatylus.

(A–C) PEFO 34045/19, Anterior trunk vertebra in anterior (A), posterior (B), and lateral (C) views. (D–E) PEFO 34045/22, Posterior trunk vertebra in anterior (D) and lateral (E) views. Scale bar equals 1 cm. Abbreviations: b., broken designated element; bf, bone fragment; cpof, centropostzygapophyseal fossa; k, keel; nst, neural spine table; parp, parapophysis; podl, postzygadiapophyseal lamina; posdf, postzygapophyseal spinodiapophyseal fossa; posz, postzygapophysis; prez, prezygapophysis; pro, projection; sprf, spinoprezygapophyseal fossa; tp, transverse process; vb, ventral bar.

In posterior view, the postzygapophyses (best preserved in PEFO 34045/14-R) are also oriented about 45° above the horizontal. They are triangular in posterior view with a well-developed lateral postzygadiapophyseal lamina (podl). That lamina extends laterally to the diapophysis and forms a broad dorsal shelf of the transverse process in dorsal view (Fig. 12A). The shelf is wider proximally and narrows distally along the transverse process. Along the dorsal surface of the shelf, between the postzygapophyses and the neural spine is a pair of shallow postzygapophyseal spinodiapophyseal fossae (posdf).

The neural spine is short (32.3 mm) relative to the centrum height as in Desmatosuchus spurensis (MNA V9300) and Typothorax coccinarum (TTU P-9214). The spine is anteroposteriorly elongate, equal in length to the proximal portion of the neural arch, and the distal end is mediolaterally expanded (spine table). The anterior and posterior margins of the neural spine possess paired vertical spinoprezygapophyseal (sprl) and spinopostzygapophyseal (spol) laminae as in Desmatosuchus spurensis (MNA V9300).

The postzygapophyses bound deep oval spinopostzygapophyseal fossae (spof). These fossae are much taller than wide and are bounded laterally by thin, nearly vertical intrapostzygapophyseal laminae (tpol). These laminae meet medially at a thickened triangular area dorsal to the neural canal. Here the vertebra bears a strong posteriorly pointed projection that inserts into the ventral portion of the spinoprezygapophyseal fossa (sprf) just above the ventral bar. This projection is also present in Calyptosuchus wellesi (e.g., UCMP 139795). Ventrolateral to the postzygapophyses there are two deep centropostzygapophyseal fossae (cpof) in the proximal portions of the transverse processes.

The transverse processes extend laterally with a length of 81.6 mm in PEFO 34045/FF-51. However, in two of the other vertebrae (PEFO 34045/14-R; PEFO 34045/19-V) the transverse processes are directed more dorsolaterally (Figs. 13A and 13B). This difference also occurs in Stagonolepis robertsoni (Walker, 1961) in the more anteriorly positioned trunk vertebrae. Furthermore, the ventral surface of the centrum in these two vertebra (PEFO 34045/14-R; 19-V) is more constricted forming a blunt ventral ‘keel.’ The keel and the orientation of the transverse process are the only visible differences between and anterior and mid-trunk vertebrae in Scutarx deltatylus.

Posterior trunk vertebrae

The currently available material of Scutarx deltatylus includes seven posterior trunk vertebrae; three from PEFO 34045, three from PEFO 31217, and one from PEFO 34919. As in Desmatosuchus spurensis (MNA V9300; Parker, 2008a), the posterior trunk vertebrae are much more robust than the anterior and mid-trunk vertebrae (Figs. 13C, 13D and 14A–14C). Notable differences between the mid- and posterior trunk vertebrae in Scutarx deltatylus include an increase in the height of the neural spines and a lengthening of the transverse processes, which coincide with the loss of distinct parapophyses and diapophyses along the series. Furthermore, the centra become anteroposteriorly shorter than they are dorsoventrally tall (Fig. 13E). The neural spine characteristics are identical to those of the mid-trunk vertebrae with regard to the presence of the various vertebral laminae and associated fossae. An isolated posterior trunk vertebra from PEFO 31217 (Fig. 14C) shows that the prezygadiapophyseal laminae are even more strongly developed and extend farther laterally than in the more anterior trunk vertebrae. In the more posterior vertebra, the length ratio between the transverse process length (86.84 mm) and centrum width (53.26 mm) equals 1.63, thus the process is more than 1.5 times the width of the centrum. This is comparable to a ratio of 1.58 for the mid-trunk vertebrae.

Figure 14 Posterior trunk vertebrae of Scutarx deltatylus.

(A–B) PEFO 34045 in anterior (A) and dorsal (B) view. (C) PEFO 31217 in anterior view. Scale bar equals 1 cm. Abbreviations: cp, capitulum; cprf, centroprezygapophyseal fossa; diap, diapophysis; ns, neural spine; nst, neural spine table; parp, parapophysis; prdl, prezygadiapophyseal lamina; posdf, postzygapophyseal spinodiapophyseal fossa; posz, postzygapophysis; prez, prezygapophysis; sprf, spinoprezygapophyseal fossa; tb, tuberculum; tp, transverse process; vb, ventral bar.

This same vertebra from PEFO 31217 also lacks distinct diapophyses and parapophyses and a single-headed rib is fused onto the distal end of the process (Fig. 14C). This is also seen in Desmatosuchus spurensis (Parker, 2008a), Stagonolepis robertsoni (Walker, 1961), and Calyptosuchus wellesi (UMMP 13950). An isolated posterior trunk vertebra from PEFO 34045 (Figs. 13A and 13B) preserves the entire transverse processes and the associated fused ribs. However, the specimen differs from the previously described vertebra from PEFO 31217 in that the parapophysis and diapophysis are distinct and the rib is double-headed (Figs. 14A and 14B). Although the ribs and transverse processes are fused, the fusion is incomplete; gaps are present within the individual articulations and another gap is apparent between the anterior surface of the distal end of the transverse process and the medial surface of the capitulum of the rib (Fig. 14B). This suggests that several vertebrae in the posterior trunk series fuse with the ribs, and loss of a distinct parapophysis and diapophysis of the transverse process and of the tuberculum and capitulum of the dorsal ribs only occurred in the last one or two presacrals. Examination of UMMP 13950 (Case, 1932; Long & Murry, 1995) suggests that this loss occurs in the last three presacrals. In Stagonolepis robertsoni that condition occurs in the final two presacral vertebrae (Walker, 1961). There is no evidence in Scutarx deltatylus that the last presacral was incorporated into the sacrum as in Desmatosuchus spurensis (Parker, 2008a). The last presacral in PEFO 31217 also shows a distinct vertical offset in the ventral margins of the articular faces of the centra with the anterior face situated more ventrally. This is also the case in Stagonolepis robertsoni (Walker, 1961) and Desmatosuchus spurensis (Parker, 2008a).

Another posterior trunk vertebra, PEFO 34045/22 (Figs. 13D and 13E), lacks the transverse processes, but preserves other key characteristics of the posterior presacrals. Its neural spine is taller (81.94 mm) than the height of the centrum (61.24 mm), differing from the condition in the anterior and mid-trunk vertebrae where the neural spine is shorter than the centrum (Fig. 13D). This transition occurs at the beginning of the posterior trunk vertebrae series, because the specimen from PEFO 34045 with the fused ribs, but distinct rib facets (Figs. 14A and 14B), has a centrum and neural spine of equal height. PEFO 34045/22 also preserves the pointed posterior projection above the neural arch that is present throughout the trunk series (Fig. 13E).

Sacral vertebrae

A sacral vertebra, probably the second, is visible in ventral view in PEFO 31217 in articulation with the rest of the pelvis (Fig. 15). It is recognizable by the presence of a strong, broad sacral rib that expands laterally and anterodorsally to contact the posterodorsal margin of the left ilium. Unfortunately no other details are available for that specimen.

Figure 15 Photo and interpretive sketch of a partially articulated sacrum and anterior portion of the tail of Scutarx deltatylus (PEFO 31217).

Scale bar equals 10 cm. Abbreviations: ac, acetabulum, apib, anterior process of the iliac blade; cdv, caudal vertebra; dv, trunk vertebra; f, foramen; isc, ischia; l.il, left ilium; l.pu, left pubis; lo, lateral osteoderm; os, osteoderm; pos, paramedian osteoderm; r.il, right ilium; r.pu, right pubis; scv, sacral vertebra.

Caudal series

Vertebrae

Eight vertebrae occur in semi-articulation in PEFO 31217 posterior to the sacral vertebra described previously (Fig. 15). The first two are robust with thick flaring rims on the centra. The first vertebra has a length of 57.3 mm, and its anterior face is indistinguishable from the posterior face of the preceding sacral vertebra. Furthermore, the centrum is constricted which is unusual for an aetosaur, because the sacrals and anterior caudals usually have wide ventral surfaces (e.g., Desmatosuchus spurensis, MNA V9300). The vertebra in PEFO 31217 lacks a ventral groove and chevron facets. It is possible that this is a sacral vertebra that has been forced backwards, but the poor preservation of the specimen does not allow a firm determination. The second caudal vertebra (assuming the first described is from the caudal series) has a centrum length of 52.2 mm and a width of 61.6 mm, thus it is wider than long as is typical for the anterior caudals of aetosaurians (Long & Murry, 1995). The centrum is ventrally broad and a chevron is articulated to the posterior margin. The base of the caudal rib originates from the base of the neural arch, but laterally the rib is incomplete.

Two anterior caudal vertebrae are also known from PEFO 34045, which roughly correspond in morphology to the second and third caudal centra of PEFO 31217 (Figs. 16A–16F). These two vertebrae have blocky centra that are wider (flared centrum faces) than long. The ventral surfaces are broad, with a deep median trough bordered by two lateral ridges. These ridges terminate posteriorly into two posteroventrally facing hemispherical chevron facets (Figs. 16D and 16E). The articular faces of the centra are round in anterior and posterior views, and in lateral view these faces are offset from each other (Fig. 16F). The ventral margin of the posterior face is situated much farther ventrally than that of the anterior face, as is typical for aetosaurs (e.g., Desmatosuchus spurensis, MNA V9300). Although the neural spines are missing, it is apparent that the neural arch complex was much taller than the height of the centrum (Fig. 16C). The neural canal is oval with a taller dorsoventral axis.

Figure 16 Anterior caudal vertebrae of Scutarx deltatylus (PEFO 34045).

(A–D) anterior caudal in posterior (A), anterior (B), lateral (C), and ventral (D). (E–F) Anterior caudal vertebra in ventral (E) and lateral (F). Scale bar equals 1 cm. Abbreviations: b., broken designated element; cf, chevron facet; cr, caudal rib; gr, ventral groove; posz, postzygapophysis; prez, prezygapophysis; spof, spinopostzygapophseal fossa; sprf, spinoprezygapophyseal fossa.

The pre- and postzygapophyseal stalks are thickened and the facets are closely situated medially. They are oriented at about 30° from the horizontal. The neural arch is directed posterodorsally and the postzygapophyses project posteriorly significantly beyond the posterior centrum face (Fig. 16C). The caudal vertebrae lack diapophyseal and zygapophyseal laminae, but spinozygapophyseal fossae occur between the prezygapophyses (Figs. 16A and 16B). The caudal ribs are fully fused to the centrum. They are anteroposteriorly broad and dorsoventrally thin with flat dorsal surfaces and buttressed ventral margins. The ribs are directed slightly posteriorly and laterally they arc ventrally (Figs. 16A–16C). Unfortunately their lateral extent is unknown.

The third and fourth caudal vertebrae in PEFO 31217 are longer than wide, with the centrum narrowing mediolaterally and with reduced flaring of the rims as in the previous vertebrae (Fig. 15). The posteroventral margins possess chevron facets. The caudal ribs are broad, flat, and were elongate, as in Desmatosuchus spurensis (MNA V9300), even though the distal ends are not preserved. The third centrum has a length of 56.4 mm and the fourth has a length of 56.4 mm. Details of the neural arches and spines are buried in the block and irretrievable by mechanical preparation.

The fifth and sixth caudal vertebrae are mostly concealed beneath armor, bone fragments, and what are probably the eighth and ninth caudal vertebrae. Only the left caudal ribs are apparent, jutting out of the block. They are dorsoventrally flat and laterally elongate, typical for aetosaurs, but they are poorly preserved and no other details are apparent.

The anterior face of what is probably the seventh caudal vertebra is visible underneath matrix and an osteoderm about six centimeters behind where the sixth caudal vertebra is buried in the block, breaking the line of articulation. The neural canal is prominent on this vertebra and what is visible of the neural arch shows that it was tall. The centrum is amphicoelous and mediolaterally constricted. The ventral surface consists of a median ventral groove bounded laterally by two sharp ridges. The ridges would terminate posteriorly with the chevron facets, but the relevant area is obliterated. A vertebra from approximately the same position is preserved in PEFO 34919 (Figs. 17A–17C) and provides more details.

Figure 17 Mid-caudal vertebrae of Scutarx deltatylus.

(A–C) anterior mid-caudal vertebra (PEFO 34919) in lateral (A), anterior (B), and posterior (C) views. (D) posterior mid-caudal vertebra (PEFO 34045) in lateral view. Scale bar equals 1 cm. Abbreviations: cf, chevron facet; cr, caudal rib; ns, neural spine; prez, prezygapophysis; posz, postzygapophysis.

The centrum is much longer than wide (57–30 mm), mediolaterally compressed, and grooved ventrally. Its rims flare minimally, but the articular faces are deeply concave (Figs. 17B and 17C). The neural arch is dorsoventrally shorter than in the more anteriorly positioned caudal vertebrae, but the neural spine was certainly tall in this position as well (Fig. 17B). The zygapophyses are reduced and each pair is closely situated medially. The postzygapophyses do not project far posteriorly. The caudal rib is situated anteroventrally on the neural arch. It is broad and flat, extends laterally (∼50 mm), and is slightly arcuate in anterior view (Fig. 17B).

What are probably the eighth and ninth caudal vertebrae are well-preserved at the edge of the block in PEFO 31217 (Fig. 15). The centra are much longer than wide. The ninth centrum has a length of 66.3 mm and a width of 40.2 mm. The lateral faces of the centrum are concave and, as on the preceding centra, the ventral face is narrow with a deep median groove terminating at the chevron facets. The neural arches and spines are complete and tall, with a height of 100.9 mm in the eighth vertebra and 98.4 mm in the ninth. The neural spines are tall and roughly triangular in lateral view, with an anteroposteriorly broad base and tapering distally. The zygapophyses are closely situated medially and extend anteriorly and posteriorly beyond the articular faces of the centra. The caudal ribs are greatly reduced in lateral length.

An isolated vertebra from PEFO 34045 represents the mid-caudal series (Fig. 17D). The centrum is longer than tall (65–35 mm) and mediolaterally compressed. Its articular faces are deeply concave and oval with the longest axis situated dorsoventrally. The neural arch is dorsolaterally reduced and mediolaterally compressed. The caudal ribs are greatly reduced and eroded. The neural spine is elongate, but its full dorsal extent is unknown (Fig. 17D).

Chevrons

Only half of a single chevron and part of the head of a second are preserved in PEFO 34045 (Figs. 18A and 18B). A few are smashed beneath other elements in PEFO 34919 and a badly preserved chevron is present beneath the second caudal vertebra of PEFO 31217. Although the details are poor the latter suggests, in accordance with the lack of facets on the first caudal vertebra of PEFO 31217, that chevrons started on the second caudal centrum. This is different from the condition in Desmatosuchus spurensis, in which they first appear on the third caudal centrum (Parker, 2008a), but similar to the condition in Typothorax coccinarum (Heckert et al., 2010). The two preserved chevrons in PEFO 34045 are of the ‘slim’ elongate type and, therefore, from the anterior portion of the tail (Parker, 2008a).

Figure 18 Chevrons and ribs of Scutarx deltatylus.

(A–B) partial anterior chevrons from PEFO 34045 in posterior view; (C–D) left trunk rib from PEFO 34045 in posterior (C) and anterior (D) views. (E) close-up view of head of trunk rib from PEFO 34045. (F) paired gastral ribs from PEFO 34616. Scale bars equals 1 cm. Abbreviations: cp, capitulum; fo, fossa; gr, groove; tb, tuberculum.

Ribs

Presacral

No cervical ribs are preserved in any of the specimens, but trunk ribs are common. The sacral and caudal ribs have been described above along with their associated vertebrae. The anterior and mid-trunk ribs are double-headed (Figs. 18C and 18D). They extend laterally for the first quarter of their total length and then turn sharply ventrolaterally, are straight for half of the total length, and then gently turn more ventrally. Proximally the rib body is oval in cross-section, becoming ovate and then flattened more distally; it is broadest at the point of the sharp ventrolateral turn.

The capitulum is oval in cross-section, with a sharp posterior projection. The capitulum and tuberculum are separated by 44 mm. The dorsal surface of the neck is marked by a transverse groove that terminates at a fossa on the proximal surface of the tuberculum (Fig. 18E). That groove probably hosted the ventral portion of the vertebrarterial canal as in Alligator (Reese, 1915). A thin flange of bone originates on the dorsal surface of the tuberculum and extends laterally, becoming confluent with the rib body just lateral to the ventrolateral hook. That flange forms a deep, elongate groove along the posterodorsal surface of the rib. Dorsally the rib is flattened and forms a thin anterior blade. The posterior-most ribs are single headed and fused with the transverse processes of the trunk vertebrae (Fig. 14C).

Gastralia

It has been suggested that aetosaurians lack gastralia (Nesbitt, 2011), but they are present in Typothorax coccinarum and Stenomyti huangae (Heckert et al., 2010; Small & Martz, 2013). In Typothorax coccinarum (e.g., NMMNH P-56299), the gastralia are preserved in the posteroventral portion of the thoracic region, are medially fused and laterally elongate. The gastralia of Stenomyti huangae (DMNH (DMNS) 60708) are presently undescribed. A single gastralia set is preserved in PEFO 34616 demonstrating that they were present in Scutarx deltatylus as well (Fig. 18F). This set consists of incomplete but medially fused ribs with a short anterior projection.

Appendicular girdles

Scapulocoracoid

The left scapulocoracoid is preserved in PEFO 31217; unfortunately the coracoid is covered by osteoderms that cannot be removed without causing significant damage, so only the dorsal-most portion of the coracoid, where it sutures to the scapula, is visible. In lateral view the general outline of the scapula of PEFO 31217 (Fig. 19A) strongly resembles the scapulocoracoid of Stagonolepis robertsoni (Walker, 1961: Fig. 12A); although it is broader anteroposteriorly. The proximal end is expanded anterolaterally with the posterior projection situated more dorsally than the anterior projection. The posterior projection has a rounded posterior margin, as in Stagonolepis robertsoni (Walker, 1961) differing from the pointed projection in Stagonolepis olenkae (ZPAL AbIII/694). The anterior projection is poorly preserved but appears to be pointed as in Stagonolepis robertsoni (Walker, 1961). The scapular blade is gently bowed medially and the posterior edge is straight except for a slight posterior projection (the triceps tubercle) about 62 mm above the glenoid lip (Fig. 19A). The anterior edge of the blade is straight for most of its length until it strongly flares anteriorly, forming a prominent deltoid ridge (= acromion process; Brochu, 1992; Martz, 2002). Below this there is a prominent foramen, although its anterior edge is broken away. Likewise the ventral margin of the posterior edge of the scapular blade strongly flares posteriorly forming the supraglenoid buttress. The glenoid facet opens posteriorly. Laterally there is a sharp ridge, which probably represents deformation and crushing along the scapulocoracoid suture.

Figure 19 Left scapulocoracoid of Scutarx deltatylus of PEFO 31217 in lateral view.

Scale bar equals 10 cm. Abbreviations: ap, acromion process; cor, coracoid; fm, foramen; ost, osteoderms; sgb, supraglenoid buttress; tt, triceps tubercle.

Ilium

Ilia are preserved in PEFO 34919 (right ilium; Fig. 20) and PEFO 31217 (both ilia; Figs. 15 and 21). When articulated the ilia of Scutarx deltatylus were oriented so that the acetabula faced ventrally as in some aetosaurs such as Aetosauroides scagliai (PVL 2073) and Typothorax coccinarum (PEFO 33980); however, to avoid confusion in this description, the anatomical directions will be provided as if the reader is viewing the ventral surface as the lateral surface (see Figs. 20A and 20B). The right ilium of PEFO 34919 is nearly complete, missing only a portion of the anterior margin of the acetabulum (Figs. 20A and 20B). As usual for the bones from this specimen, the ilium is covered with a thin layer of weathered hematite that cannot be removed without damaging the underlying bone. The iliac blade is complete, with a length of 196 mm and a mid-height of 66.8 mm. The ‘dorsal’ margin of the iliac blade is mediolaterally narrow, expanding anteriorly so that the dorsal margin of the anterior process is thicker and more robust than the rest of the blade. The anterior portion of the iliac blade is triangular in lateral view, and does not extend anteriorly beyond the edge of the pubic peduncle as in Stagonolepis robertsoni (Walker, 1961). There is a prominent recess on the dorsal surface between the supraacetabular crest and the posterior iliac blade (Fig. 20A) that appears to be unique to Scutarx deltatylus.

Figure 20 (A–B) right ilium of PEFO 34919 in ‘lateral’ and ‘medial’ views (see text for discussion regarding anatomical direction of the ilium).

Scale bar equals 1 cm. Abbreviations: ac, acetabulum; apib, anterior process of the iliac blade; fm, foramen; ip, ischiadic peduncle; pp, pubic peduncle; ppib, posterior process of the iliac blade; re, recess; sac, supraacetabular crest; sh, shelf; sras, sacral rib attachment surfaces.

Figure 21 Close-up of pelvis of Scutarx deltatylus (PEFO 31217).

Scale bar equals 5 cm. Abbreviations: ac, acetabulum, apib, anterior process of the iliac blade; f, foramen; isc, ischia; l.il, left ilium; l.pu, left pubis; r.il, right ilium; r.pu, right pubis.

The dorsoventral height of the posterior portion of the iliac blade diminishes posteriorly, terminating in a point. From there the posteroventral margin slopes anteroventrally into a curving posterior margin that distally hooks posteriorly and thickens to form the ischiadic peduncle. The posterior projection of the ischiadic peduncle is proportionally larger and more pointed than the same structure in Aetosauroides scagliai (PVL 2073) and Stagonolepis robertsoni (NHMUK R4789a), and more like that of TMM 31100-1, which represents a desmatosuchine aetosaurine (unpublished data). The ventral margins of the pubic and ischiadic peduncles meet at an angle of 90° ventral to the acetabulum, with the ilium contributing to the majority of the acetabulum. In ventral view the margins of the peduncles are comma-shaped, thinning into the ventral margin of the broadly concave acetabulum. The medial side of the acetabulum is smooth and slightly convex.

Dorsal to the iliac neck, the medial side of the posterior portion of the iliac blade bears a prominent ventral ridge that forms a shelf for sacral rib articulation (Fig. 20B). The rib scar is situated just above the ridge and forms a concave sulcus that extends anteriorly to just dorsal to the anterior margin of the neck.

Both ilia are present in PEFO 31217 as portions of a complete sacrum. Of the two the left is the better preserved. The acetabula are deeply concave and oriented ventrally (Figs. 15 and 21). Originally this was thought to be the result of crushing of the pelvis; however, the acetabula are oriented ventrally in many other uncrushed aetosaurian specimens including Aetosauroides scagliai (Heckert & Lucas, 2002), the holotype of Typothorax antiquus (Lucas, Heckert & Hunt, 2003), and Typothorax coccinarum (Heckert et al., 2010). The supraacetabular ridge in these ilia is strong, but not as strong as in rauisuchids. As in PEFO 34919, there is a deep fossa/recess on the dorsal surface between the supraacetabular ridge and the posterior portion of the iliac blade, a condition that appears to be autapomorphic for this taxon. That fossa is bordered posteroventrally by the thickened margin of the neck, a feature which is ventrally confluent with the ischiadic peduncle. The left iliac blade measures 188.6 mm in length and 67.4 mm in height, producing a relatively tall iliac blade. The posterior portion of the iliac blade has a posterior margin that projects well beyond the iliac peduncle. The extent of the ventral portions of the ilia is hard to determine because they are indistinguishably fused to the ischia and pubes; however, the left acetabulum is more or less rounded, 116.5 mm tall and 111 mm wide.

Ischium

The left ischium and part of the right are present, but poorly preserved (Figs. 15 and 21). The ischium consists of the main body with a sharp, rounded acetabular rim, and an elongate posterior process. The upper margin of the posterior process slopes gradually from the posterior margin of the ischiadic peduncle, and the entire ischium measures 183 mm in length. The anteroventral margin is flat where the two ischia are fused, forming a wide, slightly concave ventral shelf. Overall the ischium is similar to that of other aetosaurians such as Stagonolepis robertsoni (Walker, 1961), but lacks the prominent ventral kink found in Desmatosuchus spurensis (MNA V9300; Parker, 2008a).

Pubis

Both pubes are present and in articulation with the pelvis, although they are moderately distorted by crushing and were damaged by weathering before collection (Figs. 15 and 21). The body of the pubis consists of an elongate, narrow rod that curves anteroventrally and expands medially into two broad sheets of bone that meet in a median symphysis. This pubic apron is convex anteriorly and concave posteriorly. It is dorsoventrally short, barely extending past the ventral margin of the puboischiadic plate, more like the condition in Typothorax coccinarum (Long & Murry, 1995) rather than the extremely deep pubic apron found in Desmatosuchus spurensis (MNA V9300). Two distinct oval foramina pierce the pubic apron in the proximal part of the element. The bone is broken around the more anterior foramen of the right pubis, but it is clear that it was the larger of the two openings (Fig. 21). Two pubic foramina are also described for Stagonolepis robertsoni (Walker, 1961), and the upper (anterior) opening considered homologous to the single foramen found in other aetosaurs (e.g., MNA V9300, Desmatosuchus spurensis). The distal ends of the pubes are shaped like elongate commata, narrow and curving into the symphysis (Fig. 21), different from the strong, knob-like projections (pubic boots) found in Desmatosuchus spurensis (MNA V9300).

Osteoderms

Paramedian osteoderms

Cervical

Cervical osteoderms are present in PEFO 31217, PEFO 34045, and PEFO 34616. All of the osteoderms are wider than long (w/l ratio of 1.85). The cervical osteoderms are dorsoventrally thick with well-developed anterior bars (sensu Long & Ballew, 1985), which bear prominent anteromedial projections. The lateral edges are strongly sigmoidal, and lack anterolateral projections (Figs. 22A, 22C and 23A).

Figure 22 Cervical and dorsal trunk paramedian osteoderms of Scutarx deltatylus from PEFO 34045.

(A–B) left mid-cervical osteoderm in dorsal (A) and posterior (B) views. (C–D) right mid-cervical osteoderm in dorsal (C) and posterior (D). (E–F) left (E) and right (F) dorsal trunk osteoderms in dorsal view. (G–I) left (G, H) and right (I) dorsal trunk osteoderms in dorsal (G, I) and posterior (H) views. (J–K) left (J) and right (K) dorsal trunk osteoderms in dorsal view. (L–M) posterior dorsal trunk osteoderm in dorsal (L) and posterior (M) views. Scale bar equals 1 cm. Abbreviations: ab, anterior bar; alp, anterolateral process; amp, anteromedial process; anp, anterior process; de, dorsal eminence; trp, triangular protuberance.

Figure 23 Holotype paramedian osteoderms of Scutarx deltatylus from PEFO 34616.

(A) posterior cervical osteoderm in dorsal view. (B–C) right dorsal trunk paramedian osteoderm in dorsal (B) and posterior (C) views. (D–E) partial right dorsal trunk paramedian osteoderm in dorsal (D) and posterior (E) views. Note the prominence of the triangular protuberance in the posterior views. Scale bar equals 1 cm. Abbreviations: ab, anterior bar; alp, anterolateral process; amp, anteromedial process; de, dorsal eminence; trp, triangular protuberance.

The dorsal surface is relatively featureless, with the ornamentation poorly developed. The dorsal eminence is low, broad, and mounded, contacting the posterior plate margin (Figs. 22A and 22C). The eminence is also slightly offset medially, closer to the midline margin. The characteristic triangular protuberance that diagnoses Scutarx deltatylus is present in the posteromedial corner of the osteoderm, but is greatly reduced in area (Figs. 22A, 22C, 22D and 23A). In the cervical paramedian osteoderms the shape of that protuberance is more of a right triangle than the equilateral triangles found in the trunk series (see below).

In posterior view, the osteoderms are gently arched (Figs. 22B and 22D). The median margins are sigmoidal in medial view and dorsoventrally thick as is typical for aetosaurians. Scutarx deltatylus lacks the ‘tongue-and-groove’ lateral articular surfaces present in Desmatosuchus (e.g., MNA V9300) and Longosuchus meadei (TMM 31185-84b).

The more posterior cervical paramedian osteoderms are similar, but increase in width (w/l ratio of 2.05) and lack the strongly sigmoidal lateral margin. The margin is still sigmoidal but bears a strong anterolateral projection (Fig. 23A). Moreover, the anterior and posterior plate margins are gently curved anterolaterally. In posterior view, these osteoderms have a lesser degree of arching and are dorsoventrally thinner than the more anteriorly situated osteoderms. The dorsal eminence is strongly offset medially and slightly more developed, becoming raised and more pyramidal in shape, although this could be an individual variation (see description of caudal paramedian osteoderms).

Trunk

The osteoderm transition between the cervical and trunk series is difficult to identify, but anterior dorsal trunk osteoderms are considered here to have higher width/length ratios and be dorsoventrally thinner than the cervical paramedian osteoderms. Furthermore, the triangular protuberance is more equilateral. However, it is difficult to differentiate these osteoderms from those of the anterior caudal region.

Osteoderms with the maximum width/length ratio (2.72/1) are found in the mid-trunk region. They bear a strongly raised anterior bar with prominent anteromedial and anterolateral projections. Prominent (greatly elongate) anterolateral projections also occur in Calyptosuchus wellesi (UMMP 7470), Adamanasuchus eisenhardtae (PEFO 34638), and Neoaetosauroides engaeus (PVL 3525). The anterolateral projections are shorter in Stagonolepis robertsoni (NHMUK 4790a) and Aetosauroides scagliai (PVL 2073). The dorsal eminence in Scutarx deltatylus is medially offset, and forms a broad, low mound. Anterior to this on the anterior bar is a prominent, pointed anterior projection. The area of the anterior bar medial to this process is ‘scalloped out,’ and as a result is deeply concave (Figs. 22E, 22F, 22J and 22K). This ‘scalloping’ of the anterior bar is a synapomorphy of aetosaurine aetosaurs, occurring throughout the clade. The length of the anterior bar decreases significantly within the arc of this concavity. The triangular protuberance is prominent and equilateral (Figs. 22E–22K, 23B and 23D).

The lateral margin is sigmoidal, and the anterior portion just posterior to the anterior bar is slightly embayed for slight overlap of the associated lateral osteoderm. In posterior view the osteoderm is only slightly arched (Fig. 22H). In what are presumed to be more posteriorly positioned osteoderms, the osteoderm is more strongly arched (Figs. 22L and 22M). The triangular protuberances are particularly visible in posterior view, extending even further dorsally that the main dorsal eminence (Figs. 23C and 23E). The ventral surface of the dorsal trunk paramedian osteoderms are smooth, with a slight embayment situated on the underside of the dorsal eminence.

The surface ornamentation of the dorsal trunk paramedian osteoderms is barely apparent in PEFO 34045, but much better developed in the other specimens. The ornament consists of pitting surrounding the dorsal eminence and radiating grooves and ridges over the rest of the surface. The ornamentation in Calyptosuchus wellesi lacks these strong radiating grooves.

There is no direct evidence for a constriction (‘waist’) in the carapace anterior to the pelvis as in Aetosaurus ferratus (Schoch, 2007), Calyptosuchus wellesi (Case, 1932), and Aetosauroides scagliai (Heckert & Lucas, 2002); however, because the lateral osteoderm shapes in Scutarx deltatylus are identical to those of Calyptosuchus wellesi, it is probable that Scutarx deltatylus also possessed a ‘waisted’ carapace although this cannot be confirmed.

Overall the paramedian cervical and trunk osteoderms of Scutarx deltatylus are similar to those of Calyptosuchus wellesi in all characteristics except for the presence of the posteromedial triangular protuberance (Fig. 24). This character must be present to differentiate Calyptosuchus wellesi and Scutarx deltatylus paramedian osteoderms and in osteoderms where this area is not preserved an alpha taxonomic assignment cannot be made.

Figure 24 Comparison of dorsal trunk paramedian osteoderms of Calyptosuchus wellesi (A) and Scutarx deltatylus (B–D) in dorsal view.

(A) MNA 2930, left osteoderm of Calyptosuchus wellesi lacking the triangular protuberance (trp). (B) UCMP 36656, right osteoderms of Scutarx deltatylus showing the triangular protuberance. (C) UCMP 126844, medial portion of left osteoderm of Scutarx deltatylus showing the triangular protuberance. (D) UCMP 35738, medial half of left osteoderm of Scutarx deltatylus showing the triangular protuberance. Scale bar equals 5 cm. Abbreviations: ab, anterior bar; alp, anterolateral process; amp, anteromedial process; de, dorsal eminence; trp, triangular protuberance.

Caudal

Like the cervical-trunk transition, the trunk-caudal transition is also difficult to determine in unarticulated aetosaurian carapaces (Parker, 2008a). The latter transition is generally characterized by reduction of osteoderm width-length ratios and greater development of the dorsal eminences (Heckert & Lucas, 2000). The extreme is found in Rioarribasuchus chamaensis, in which the barely visible dorsal eminences in the mid-dorsal region transition posteriorly to elongate, anteromedially curved spines in the anterior caudal region (Parker, 2007).

The trunk-caudal transition for Scutarx deltatylus is best preserved in PEFO 34919 in which the dorsal eminences show a marked increase in height from 16.35 mm in the mid-trunk region to 40.07 mm in the anterior dorsal caudal region. Width/length ratios across this same transition are 2.54–2.16, showing the corresponding decrease. The dorsal eminence is a tall pyramid, with a posterior vertical keel (Fig. 25). In all other respects the anterior caudal osteoderms are similar to those of the trunk region.

Figure 25 Fused semi-articulated anterior dorsal caudal paramedian and dorsal caudal lateral osteoderms of Scutarx deltatylus (PEFO 34919) in a lateral view showing extreme development of the dorsal eminences.

Scale bar equals 1 cm. Abbreviations: lo, lateral osteoderm; po, paramedian osteoderm.

Dorsal mid-caudal paramedians are relatively equal in width and length (w/l ratio = 1.08). Those osteoderms still possess the pronounced dorsal eminence (Figs. 26A–26J), as well as the anteromedial and anterolateral projections of the anterior bar. In PEFO 34045 these osteoderms are extremely thickened (Figs. 26A, 26B, 26E and 26F).

Figure 26 Dorsal caudal paramedian osteoderms of Scutarx deltatylus.

(A–B) left anterior mid-caudal osteoderm (PEFO 34045) in dorsal (A) and posterior (B) views. (C–D) right anterior mid-caudal osteoderm (PEFO 34919) in dorsal (C) and posterior (D) views; (E–F) left mid-caudal osteoderm (PEFO 34045) in dorsal (E) and posterior (F) views. (G–H) right mid-caudal osteoderm (PEFO 34919) in dorsal (G) and posterior (H) views. (I–J) left mid-caudal osteoderm (PEFO 34919) in dorsal (I) and posterior (J) views. (K–L) right posterior caudal osteoderm (PEFO 34045) in dorsal (K) and posterior (L) views. (M–N) left posterior caudal osteoderm (PEFO 34045) in dorsal (M) and posterior (N) views. Scale bar equals 1 cm. Abbreviations: ab, anterior bar; alp, anterolateral process; amp, anteromedial process; de, dorsal eminence; me, medial edge.

The posterior dorsal caudal paramedians (Figs. 26K–26N) become longer than wide (w/l ratios of 0.73 and 0.66), and the dorsal eminence is reduced to a raised, anteroposteriorly elongate keel with a posterior projection that extends beyond the posterior margin of the osteoderm. Presumably these continue until they become elongate strips of bone as in Aetosaurus ferratus (Schoch, 2007) and Typothorax coccinarum (NMMNH P56299; Heckert et al., 2010).

Lateral osteoderms

The best guide for the distribution of the lateral osteoderms is UMMP 13950, the holotype of Calyptosuchus wellesi, which preserves the posterior dorsal armor and much of the caudal lateral armor in articulation (Case, 1932). Scutarx deltatylus possesses lateral plates that are identical in shape to those of Calyptosuchus wellesi, allowing for determination of caudal and posterior dorsal osteoderms. Therefore, any lateral osteoderms falling outside of those morphotypes probably are from more anterior regions. Anterior dorsal lateral osteoderms are preserved in the articulated holotype of Aetosauroides scagliai (PFV 2073), which can be used to help assign isolated osteoderms.

Lateral osteoderms can be distinguished from paramedian osteoderms primarily by the lack of the prominent anterolateral projection. Furthermore, the anteromedial corner of the osteoderm is ‘cut-off’ and beveled for reception of the anterolateral projection of the associated adjacent paramedian osteoderm (poa; Fig. 27).

Figure 27 Lateral osteoderms of Scutarx deltatylus.

(A–B) left anterior trunk osteoderm (PEFO 34616) in dorsal (A) and posterior (B) views; (C–D) right anterior trunk osteoderm (PEFO 34045) in dorsal (C) and posterior (D) views; (E–F) right posterior mid-trunk osteoderm (PEFO 34045) in dorsal (E) and posterior (F) views; (G–H) left posterior mid-trunk osteoderm (PEFO 34045) in dorsal (G) and posterior (H) views; (I–J) right posterior trunk osteoderm (PEFO 34045) in dorsal (I) and posterior (J) views; (K–L) right anterior dorsal caudal osteoderm (PEFO 34045) in dorsal (K) and posterior (L) views; right posterior dorsal mid-caudal osteoderm (PEFO 34919) in dorsal (M) and posterior (N) views; (O–P) left dorsal mid-caudal osteoderm (PEFO 34616) in dorsal (O) and posterior (P) views. Scale bar equals 1 cm. Abbreviations: ab, anterior bar; alw, anterolateral wing; de, dorsal eminence; df, dorsal flange; mf, medial flange; poa, paramedian osteoderm articular surface.

Cervical

There are no lateral osteoderms in the material present that can unequivocally be assigned to the cervical region.

Trunk

Anterior lateral trunk osteoderms are not preserved in the holotype of Calyptosuchus wellesi, but they are preserved in Aetosaurus ferratus (Schoch, 2007). In Aetosaurus those osteoderms are strongly asymmetrical with the dorsal flanges roughly half the dimensions of the lateral flanges. Furthermore, the dorsal flanges are triangular or trapezoidal in dorsal view rather than rectangular, with a slight, medially projecting posterior tongue.

Two osteoderms from the left side in PEFO 34616 and a third from the right side in PEFO 34045 match this anatomy and are probably from the anterior portion of the carapace (Figs. 27A–27D). In addition to the features just mentioned, those osteoderms possess a distinct anterior bar. The anteromedial corner of the anterior bar is beveled for articulation with the anterolateral process of the paramedian osteoderm. The dorsal eminence of the lateral osteoderm is a prominent pyramidal boss that contacts the posterior plate margin and extends anteriorly, covering two-thirds of the osteoderm length. Surface ornamentation consists of elongate grooves and ridges radiating from the dorsal eminence. In posterior view, the osteoderms are only slightly angulated, with the angle between flanges strongly obtuse (Figs. 27B and 27D). Similarly shaped osteoderms are found in the anterior lateral trunk region of Aetosauroides scagliai (PVL 2073).

Posterior-mid trunk osteoderms (from roughly the ninth through 12th positions) are sub-rectangular with a distinct, posteromedially sloping lateral edge (Figs. 27E–27H; Case, 1932). The dorsal flange is sub-rectangular in dorsal view. The medial edge of the dorsal flange is beveled and slightly sigmoidal with a ‘cut-off’ anterior corner for the anterolateral projection of the paramedian plate. The osteoderm is moderately flexed with the lateral flange extending at about 45° relative to the dorsal flange (Figs. 27F and 27H). Both flanges are roughly the same size, although the sloping lateral edge produces a small anteromedial ‘wing’ that extends that edge a bit farther laterally and provides a trapezoidal shape for the lateral flange (alw; Figs. 27E and 27G). The dorsal eminence is pyramidal, and the degree of its development differs between specimens, from a low mound in PEFO 34045 to a distinct tall, triangular boss in PEFO 34919. On the dorsal surface a distinct anterior bar is present and the surface ornamentation consists of small pits and elongate grooves radiating from the dorsal eminence. Ventrally the osteoderms are smooth, except for longitudinal striations along the posterior margin where this margin would overlap the anterior bar of the preceding lateral osteoderm.

The posterior-most lateral trunk osteoderms (15th and 16th positions) are similar to the posterior mid-trunk osteoderms but lack the anterolateral ‘wing’ and are much more strongly flexed, enclosing an angle of approximately 90° in posterior view (Figs. 27I and 27J). They are similar to the posterior lateral trunk osteoderms in Calyptosuchus wellesi (Case, 1932).

Caudal

Caudal lateral osteoderms are more equal in dimension, and bear rectangular dorsal flanges (Figs. 27K–27P). The angle enclosed between the dorsal and lateral flanges is about 45–50° (Figs. 27L, 27N and 27P). Overall these osteoderms possess some of the same surficial features as the other osteoderms, such as an anterior bar, radial ornamentation, and a posteriorly placed dorsal eminence. However, the anterior caudal osteoderms in some specimens (e.g., PEFO 34919) possess some of the tallest dorsal eminences in the carapace (Figs. 25 and 27N). The caudal lateral osteoderms also decrease in width posteriorly (Figs. 27M and 27N). The height of the dorsal eminence is gradually reduced and becomes an elongate sharp ridge.

Ventral trunk osteoderms

Ventral trunk osteoderms are preserved in all of the PEFO specimens, including an articulated, but badly preserved, set in PEFO 31217. They consist mainly of square to rectangular osteoderms, with reduced anterior bars, no dorsal eminence and a surface ornamentation of pits and elongated pits in a radial pattern emanating from the center of the osteoderm (Figs. 28A–28F). Because no complete set is preserved the exact numbers of rows and column cannot be determined; however, they would have been overlapping as in Stagonolepis robertsoni (Walker, 1961) and Typothorax coccinarum (Heckert et al., 2010).

Figure 28 Ventral trunk and appendicular osteoderms of Scutarx deltatylus from PEFO 34616.

(A–F) square ventral osteoderms. (G) round, keeled appendicular osteoderm. (H) triangular ventral (cloacal?) osteoderm. (I) round, ornamented appendicular osteoderm. Scale bar equals 1 cm. Abbreviations: ab, anterior bar; k, keel.

Appendicular osteoderms

A few irregular, small, rounded osteoderms most likely represent appendicular osteoderms. There are two types: one featureless except for a distinct raised keel (Fig. 28G), and the other with a surface ornamentation of radial pits (Figs. 28G–28I). A triangular osteoderm (Fig. 28H) from PEFO 34616 could represent a different type of appendicular osteoderm, or it could also be an irregularly shaped osteoderm from the ventral carapace possibly from the vicinity of the cloaca (A. Heckert, 2016, personal communication).

Broken osteoderms

An interesting aspect of PEFO 34045 is the presence of many irregularly shaped osteoderms recovered with the specimen (Fig. 29). All of the edges on these osteoderms are compact bone and do not represent recent breaks. Close examination shows that these specimens are the lateral ends of dorsal paramedian osteoderms because they possess anterior bars with strong anterolateral projections and sigmoidal edges (Figs. 29A–29D). It is unclear why these osteoderms are incomplete but two possibilities exist. The first possibility is that these osteoderms were incompletely ossified. Alternatively, they were broken and then the edges healed during the life of the animal. However, there is no visible sign of pathology because the edges are smooth and the dorsoventral thickness of the osteoderms remains constant. The osteoderms are also from opposite sides of the body precluding a cause from a single injury if they are pathologic in nature. Histological examination could help determine the ontogeny of these elements. If growth rings are uniform throughout the specimen, it would demonstrate that either damage occurred at a young age or that the remainder of the element did not ossify. If the osteoderms were broken at a later ontogenetic stage and healed, then that should be reflected in the bone histology showing a disruption in the growth rings, or establishment of new rings along the broken edge.

Figure 29 Incompletely formed trunk paramedian osteoderms from PEFO 34045.

(A–B) right osteoderms in dorsal view; (C) left osteoderm in dorsal view; (D) right osteoderm in dorsal view. Scale bar equals 1 cm. Abbreviations: ab, anterior bar; alp, anterolateral process.

Discussion

Scutarx deltatylus exemplifies the importance of utilizing a detailed apomorphy-based approach to differentiate Late Triassic archosauromorph taxa (e.g., Nesbitt, Irmis & Parker, 2007; Nesbitt & Stocker, 2008; Stocker, 2010). The material here referred to Scutarx deltatylus was originally assigned to Calyptosuchus wellesi (Long & Murry, 1995; Parker & Irmis, 2005; Martz et al., 2013), which was differentiated from Stagonolepis robertsoni by the presence of the triangular protuberance on the paramedian osteoderms (Martz et al., 2013). However, reexamination of the holotype of Calyptosuchus wellesi (UMMP 13950) as well as referred material from the Placerias Quarry of Arizona shows that material of Calyptosuchus wellesi actually lacks the triangular protuberance. Moreover, the skull of Scutarx deltatylus possesses characters of the braincase (e.g., foreshortened parabasisphenoid) that are more similar to Desmatosuchus than to other aetosaurians that are similar to Stagonolepis. Unfortunately, the skull of Calyptosuchus wellesi is still mostly unknown. The Placerias Quarry contains a number of isolated aetosaurian skull bones (most notably basicrania), with differing anatomical characteristics, but none of these can be referred with certainty to Calyptosuchus wellesi (Parker, 2014). Nonetheless, prior to the discovery of the skull of Scutarx deltatylus, Calyptosuchus wellesi was assumed to have a skull more like that of Stagonolepis robertsoni and Aetosauroides scagliai (i.e. with an elongate parabasisphenoid). That assumption can no longer be maintained. A phylogenetic analysis (Parker, 2016) recovers Scutarx deltatylus as the sister taxon to Adamanasuchus eisenhardtae and forming a clade with Calyptosuchus wellesi. The unnamed clade formed by these three taxa is the sister taxon of Desmatosuchini (Parker, 2016) within Desmatosuchinae (Fig. 30). The presence of a aetosaurian with armor similar to Stagonolepis robertsoni (sensu Heckert & Lucas, 2000), but with a skull more like that of desmatosuchins provides further support that certain characteristic of the armor that were once used to unite taxa, such as paramedian osteoderm ornamentation (Heckert & Lucas, 2000; Long & Ballew, 1985; Long & Murry, 1995), may have wider distributions across Aetosauria than previously recognized (Parker, 2008b; Desojo & Ezcurra, 2011; Small & Martz, 2013; Heckert et al., 2015).

Figure 30 Time-calibrated phylogeny of the Aetosauria showing estimated ranges of taxa in the Triassic stages and associated vertebrate biozones.

The Adamanian biozone is highlighted in blue.

Implications for Late Triassic vertebrate biochronology

The holotype and all of the referred specimens of Scutarx deltatylus were originally assigned to Calyptosuchus wellesi (Long & Murry, 1995; Martz et al., 2013; Parker & Irmis, 2005; Parker & Martz, 2011), a proposed index taxon of the Adamanian biozone (Parker & Martz, 2011), which is earliest Norian in age (Irmis et al., 2011). However, all of the recognized specimens of Scutarx deltatylus originate only from the Adamanian portion of the Sonsela Member of the Chinle Formation and the middle part of the Cooper Canyon Formation of Texas (Martz et al., 2013; Parker & Martz, 2011). The reassignment of this material restricts the stratigraphic range of Calyptosuchus wellesi to the Bluewater Creek and Blue Mesa members of the Chinle Formation as well as the Tecovas Formation of Texas (Heckert, 1997; Long & Murry, 1995), which are stratigraphically lower than the Sonsela Member and middle part of the Cooper Canyon (Martz et al., 2013).

It has been suggested that the Adamanian biozone (sensu Parker & Martz, 2011) could possibly be subdivided into sub-zones (Martz et al., 2013). That hypothesis was supported by a list of Adamanian taxa of the Chinle Formation that noted which are known solely from the Blue Mesa Member and which are known only from the lower part of the Sonsela Member. The list of taxa shared by both units is small and consists of Placerias hesternus (a dicynodont synapsid), the archosauromorph Trilophosaurus dornorum, the poposaurid Poposaurus gracilis, a paratypothoracin aetosaur similar to Tecovasuchus chatterjeei, and Calyptosuchus wellesi (Martz et al., 2013). The reassignment of the Sonsela material previously placed in Calyptosuchus wellesi to Scutarx deltatylus further reduces that list. Scutarx deltatylus also occurs in the upper Adamanian Post Quarry of Texas, which contains taxa elsewhere only found in the lower part of the Sonsela Member (e.g., Desmatosuchus smalli, Trilophosaurus dornorum, Typothorax coccinarum, Paratypothorax sp.; Martz et al., 2013). Thus, Scutarx deltatylus can presently be considered an index taxon of the upper part of the Adamanian biozone, which is presently considered to be middle Norian in age (Fig. 30; Irmis et al., 2011).

Hunt, Lucas & Heckert (2005) previously divided the Adamanian biozone into older and younger parts, respectively called the St. Johnsian and Lamyan sub-biochrons. Index taxa of the Lamyan are the aetosaur Typothorax antiquus (=Typothorax coccinarum) and the pseudopalatine phytosaur Machaeroprosopus (Hunt, Lucas & Heckert, 2005). However, the lowest known occurrence of Machaeroprosopus would represent the base of the Revueltian biozone (Martz & Parker, in press), thus the Lamyan would be Revueltian in age and not represent a subdivision of the Adamanian (Heckert, 2006; Parker, 2006). Accordingly I leave any proposed subdivisions presently unnamed.

Conclusions

Scutarx deltatylus is a new taxon of aetosaurian from the middle Norian (late Adamanian) of the American Southwest, based on material that was originally assigned to Calyptosuchus wellesi. This taxon is known from several carapaces and includes rare skull material from western North America. Scutarx deltatylus differs from all other aetosaurians in the presence of a raised triangular boss in the posteromedial corner of the presacral paramedian osteoderms, a dorsoventrally thickened skull roof, and an anteroposteriorly shortened parabasisphenoid. A phylogenetic analysis places it as the sister taxon of Adamanasuchus eisenhardtae near the base of Desmatosuchinae (Parker, 2016). Scutarx deltatylus appears to have utility as an index taxon for the late Adamanian biozone.

Much of this manuscript was a part of a doctoral dissertation submitted to the University of Texas at Austin. Reviews of that earlier version were provided by Tim Rowe, Chris Bell, Sterling Nesbitt, and Hans-Dieter Sues. Reviews by Sarah Werning, Julia Desojo, and Andrew Heckert greatly improved the manuscript and figures. Thank you to the management and staff of Petrified Forest National Park (PEFO) for their support of this project. For fieldwork assistance at the Petrified Forest, I thank Daniel Woody, David Gillette, Sue Clements, Dan Slais, Randall Irmis, Sterling Nesbitt, Jeff Martz, Michelle Stocker, Raul Ochoa, Lori Browne, Chuck Beightol, Rachel Guest, Matt Smith, and Kenneth Bader. Raul Ochoa discovered the type specimen of Scutarx deltatylus. I appreciate the assistance provided by the Maintenance Division staff of PEFO in the final collection of many of these specimens. Preparation of PEFO specimens was completed by Pete Reser, Matt Brown, Matt Smith, and Kenneth Bader. All specimens were collected under a natural resources permit from the National Park Service.

Access to specimens under their care was provided by T. Scott Williams and Matt Smith (PEFO); Pat Holroyd, Mark Goodwin, and Kevin Padian (UCMP); David and Janet Gillette (MNA); Julia Desojo (MACN); the late Jaime Powell (PVL); Ricardo Martinez (PVSJ); Sandra Chapman, Lorna Steel, and David Gower (NHMUK); Lindsay Zanno and Vince Schneider (NCSM); Sankar Chatterjee and Bill Mueller (TTU P); Matthew Carrano (USNM); Tony Fiorillo and Ron Tykoski (DMNH (PMNH)); Alex Downs (GR); Charles Dailey and Dick Hilton (Sierra College); Tim Rowe, Lyndon Murray, Matt Brown, and Chris Sagebiel (VPL).

This is PEFO Paleontological Contribution number 39.

Institutional abbreviations

DMNH (PMNH) Perot Museum of Natural History, Dallas, Texas, USA

DMNH (DMNS) Denver Museum of Nature and Science, Denver, Colorado, USA

MCZD Marischal College Zoology Department, University of Aberdeen, Aberdeen, Scotland, UK

NCSM North Carolina State Museum, Raleigh, North Carolina, USA

NHMUK The Natural History Museum, London, United Kingdom

NMMNH New Mexico Museum of Natural History and Science, Albuquerque, New Mexico, USA

MNA Museum of Northern Arizona, Flagstaff, Arizona, USA

PEFO Petrified Forest National Park, Petrified Forest, Arizona, USA

PFV Petrified Forest National Park Vertebrate Locality, Petrified Forest, Arizona, USA

PVL Paleontología de Vertebrados, Instituto ‘Miguel Lillo,’ San Miguel de Tucumán, Argentina

PVSJ División de Paleontologia de Vertebrados del Museo de Ciencias Naturales y Universidad Nacional de San Juan, San Juan, Argentina

TMM Texas Vertebrate Paleontology Collections, University of Texas, Austin, Texas, USA

TTU P Museum of Texas Tech, Lubbock, Texas, USA

UCMP University of California, Berkeley, California, USA

UMMP University of Michigan, Ann Arbor, Michigan, USA

USNM National Museum of Natural History, Smithsonian Institution, Washington, D.C., USA

VPL Vertebrate Paleontology Lab, University of Texas at Austin, Austin, Texas, USA

YPM Yale Peabody Museum of Natural History, New Haven, Connecticut, USA

ZPAL Institute of Paleobiology of the Polish Academy of Sciences in Warsaw, Warsaw; Poland.

Additional Information and Declarations

Competing Interests

Author Contributions

Data Deposition

The author declares that he has no competing interests.

William G. Parker conceived and designed the experiments, performed the experiments, analyzed the data, contributed reagents/materials/analysis tools, wrote the paper, prepared figures and/or tables, reviewed drafts of the paper.

The following information was supplied regarding data availability:

The research in this article did not generate any raw data.

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
