# Peer review of "Osteology of the Late Triassic aetosaur Scutarx deltatylus (Archosauria: Pseudosuchia)"

_PeerJ, doi:10.7717/peerj.2411_

## Round 0.1 · original submission · Minor Revisions

Dear author,

I have accepted the reviewers' decision of 'minor revisions'. Both reviewers have mentioned that there are some language issues which will require fixing. Please note that PeerJ does not provide a full proof-stage language review, as such any changes need to be made by the author prior to acceptance.

Both reviewers also mention possible improvements to the figures that should be considered.

Once again, thank you for submitting your manuscript to PeerJ and I look forward to receiving your revision.

·

Basic reporting

In general the paper is well written, organized and very interesting, and I would recommend publication in Peerj.
It is an important contribution to the anatomy of a new aetosaur, mainly by the description of a partial skull (braincase), and postcraneal elements in detail.

I don’t have any major comments or remarks to manuscript. All figures are well prepared, clear and useful for comparison with information in text. I have just a few minor comments and corrections (see attached manuscript) and below. As I am not a native speaker, I did not do important language corrections.

Experimental design

No comments

Validity of the findings

No comments

Additional comments

Dear Author, I am happy with this paper, well writing and many anatomical descriptions. please, see my few important comments below and suggestions on the pdf.

-Could you indicate the specimen number (e.g. PEFO 34616, PEFO 34035, 34919) in the Map of Fig. 1? It is better to follow all the specimen numbers.

-Could you mention at the end of the Introduction section the aim of the present contribution? It is clearly not mentioned.

-Regards the section of Naming Conventions for Aetosaurian Osteoderms, the author suggests to use PLASTON for ventral amor. This term is only apply for turtle and not homologous with aetosaur ventral osteoderms, because the plastron also include appendicular elements (e.g. pectoral and pelvic girdle). I recommend to the author read a couple of paper about the plastron origin to check its definition:

Lyson et al 2013. Homology of the enigmatic nuchal bone reveals novel reorganization of the shoulder girdle in the evolution of the turtle shell. EVOLUTION & DEVELOPMENT 15:5, 317–325 (2013) DOI: 10.1111/ede.12041

Clark et al 2001. Evidence for the Neural Crest Origin of Turtle Plastron
Bones. genesis 31:111–117 (2001)

Gilbert et al. 2007. The contribution of neural crest cells to the nuchal bone and
plastron of the turtle shell. Integrative and Comparative Biology, volume 47, number 3, pp. 401–408 doi:10.1093/icb/icm020

I suggest to the author include a figure with all the skull element known for Scutarx deltatylus, even do indicate in Fig. 3 the gular and appendicular osteoderms described. Also, it would be better to incorporate a Table with measurements of each element.
The description of the Basioccipital/Parabasisphenoid should be enrich with more comparison of aetosaur taxa with braincase were preserved (e.g. Neoaetosauroides, SMNS 19003, Aetosaurus, Longosuchus)

·

Basic reporting

Please see the annotated manuscript (attached). There are numerous minor typographical and other errors. While generally well-written, these detract from the message. Most of these can be readily found on the annotated version of the manuscript attached.

Key comments: p. 7. That “age” is, of course, a maximum depositional age, and should be identified as such.

p.9-10, note that Heckert and Lucas (1999) explicitly described the paramedian and lateral armor in terms of “columns” and “rows.” Personally I think “thoracic,” a term used extensively for amniotes, is better than “trunk,” which the author notes used in amphibians, but I can live with it. Similarly, Heckert and Lucas (1999, 2000) have noted the increased size of caudal dorsal eminences (see p. 44).

I dislike use of the word “represents” for anatomical features, I think “comprises” or other words are better.

Given the extensive description of the vertebrae, which probably deserve greater treatment than they’ve traditionally been given in the past, the discussion of the osteoderms appears rushed. It generally lacks the extensive comparisons that were found in the skull section, even though there are many more aetosaurs known only from osteoderms (or at least not known from skulls) and that every author, including Parker, uses osteoderm features to distinguish many taxa. Indeed, the ornamentation pattern of the paramedians is barely mentioned and not convincingly illustrated.

p. 49 Heckert et al. (2015) also noted that there is some form of “mosaic” evolution among aetosaurs, as Gorgetosuchus has some, but not all, “Desmatosuchine” characters.

Heckert, A. B., Schneider, V. P., Fraser, N. C., and Webb, R. A., 2015, A new aetosaur (Archosauria, Stagonolepididae) from the Upper Triassic Pekin Formation, Deep River Basin, North Carolina, U.S.A. and its implications for early aetosaur evolution: Journal of Vertebrate Paleontology, v. 35, no. 1, p. e88131; 13 p.

My principal concerns regard the figures:

Most of the line art is rather ambiguous. I understand that the material is not well preserved, but the line art often has extensive areas that are not labeled and not really clear what they represent. This is particularly true of figures 4, 5 and 14

There are some minor disagreements between the text and the figures regarding labeling. See fig. 8 and the vertebral figures (Fig. 11) as an example.

The scales of the figures are disconcertingly variable. For example, Figure 7 is much smaller than Figure 6. Rotating Figure 7 90˚ would allow it to be a similar size and much more informative.

Figure 8 seems to show that the supraoccipital is excluded from the foramen magnum, contra the text on p. 19.

Figure 12 is entirely too small; I think it would be better if it just had A-F. Note that the “A” is a different size than all other labels.

I think that, to better illustrate the pelvis, a blow-up view of just the pelvic portions of
Figure 14 would be extremely helpful. This figure desperately needs additional shading or some way of discerning what is morphology and what is not. The bolder lines that, I think, delineate holes, are insufficient.

Figure 14 should also have the caudal vertebrae numbered to match the description in the text to help the reader.

Some sort of patterning or shading to show holes, matrix, the jacket, etc., is highly desirable.

The pelvic elements (Figure 18) should probably be illustrated separately and made much larger. Note that these illustrations are inferior to those of fragmentary osteoderms (Fig. 25). Given that these elements are more taxonomically and biomechanically important, they should be better illustrated.

Figs. 19-20. I understand that the figures are arranged by different specimens, but it makes comparison of morphology a little awkward. I’d prefer a figure of cervical osteoderms, then another figure with dorsal osteoderms.

Figure 21. This specimen looks extremely interesting. I would appreciate a dorsal view and perhaps an interpretive sketch of both the dorsal and lateral views.

Figure 24. I wonder if H might be a cloacal ventral osteoderm (see Heckert et al., 2010).
It’d be nice to have illustrations of the referred specimens from Texas, especially as they are used to support the author’s biostratigraphic hypothesis.

Experimental design

This is basically a sound review of the osteology of the animal. It needs some work (outlined in other sections) to be as good enough to merit publication, but otherwise it is typical paleontological work.

One of my major concerns is that the description alternates between true description of the preserved morphology and observations that would be better recorded in a "comparisons" section. See pages 22-24 as an example. This is especially true because the author sometimes notes comparisons to specimens, such as those of Stagonolepis and Aetosauroides, that have been illustrated in the literature. I do not doubt that he has personally examined those specimens, but sometimes he cites himself or other authors that have illustrated the morphology, and sometimes he doesn’t. Others have published on these specimens, and illustrated the morphology. For example, with PVL 2073 (holotype of Aetosauroides scagliai), in addition to Casamiquela (1960, 1961), both Heckert and Lucas (2002) and Desojo and Ezcurra (2011) have published illustrations of the material, but only the last is cited, and that infrequently.

Casamiquela, R. M., 1960, Notica preliminar sobre dos nuevos estagonlepoideos Argentinos: Ameghiniana, v. 2, p. 3-9.

Casamiquela, R. M., 1961, Dos nuevos estagonolepoideos Argentinos de Ischigualasto, San Juan.: Revista Asocíation Geológia Argentina, v. 16, p. 143-203.

Heckert, A. B., and Lucas, S. G., 2002, South American occurrences of the Adamanian (Late Triassic: latest Carnian) index taxon Stagonolepis (Archosauria:Aetosauria) and their biochronological significance: Journal of Paleontology, v. 76, no. 5, p. 854-863.

When it comes to biostratigraphy (e.g., p. 50), the author ignores that the Adamanian was subdivided by Hunt et al. (2005) into St. Johnsian and Lamyan divisions. They may not endorse that, and their interpretation may be different, but Lucas and colleagues have used those terms extensively.

Hunt, A. P., Lucas, S. G., and Heckert, A. B., 2005, Definition and correlation of the Lamyan: A new biochronological unit for the nonmarine late Carnian (Late Triassic): New Mexico Geological Society Guidebook, v. 56, p. 357-366.

Validity of the findings

I think the validity is fine. The author has done a competent job of describing multiple specimens that are interesting and relatively new to science.

Additional comments

I will look forward to seeing the final version.

---

## Round 0.2 · Minor Revisions

Dear author,

Reading through your track-changed document I noticed there are still some comments you added for yourself, and some text highlighted in yellow. As such, I am returning the document so you can do a final check.

I also noticed some of the newly added binomens are not in italics. Please ensure all spelling is correct, as PeerJ does not offer a full language proofing service.

---

## Round 0.3 · accepted · Accept

Dear author,

Many thanks for your revised manuscript. After reading it, I have accepted it for publication in PeerJ.

Once again, thank you for submitting your manuscript to PeerJ and I hope you will use us again as your publication venue.

If we need to clarify any details required to move the manuscript forward, then our production staff will get in touch with you. Otherwise, a proof will be forthcoming shortly for your review.

Congratulations and thank you for your submission.